# Characterization and correction of stray light in TROPOMI-SWIR

Paul J. J. Tol[1], Tim A. van Kempen[1], Richard M. van Hees[1], Matthijs Krijger[1,2], Sidney Cadot[1,3], Ralph Snel[1,4], Stefan T. Persijn[5], Ilse Aben[1], and Ruud W. M. Hoogeveen[1]

[1]SRON Netherlands Institute for Space Research, Utrecht, the Netherlands
[2]Earth Space Solutions, Utrecht, the Netherlands
[3]Jigsaw B.V., Delft, the Netherlands
[4]Science and Technology B.V., Delft, the Netherlands
[5]VSL Dutch Metrology Institute, Delft, the Netherlands

*Correspondence to:* Paul Tol (p.j.j.tol@sron.nl)

**Abstract.** The shortwave infrared (SWIR) spectrometer module of the Tropospheric Monitoring Instrument (TROPOMI), on board the ESA Copernicus Sentinel-5 Precursor satellite, is used to measure atmospheric CO and methane columns. For this purpose, calibrated radiance measurements are needed that are minimally contaminated by instrumental stray light. Therefore, a method has been developed and applied in an on-ground calibration campaign to characterize stray light in detail using a monochromatic quasi-point light source. The dynamic range of the signal was extended to more than seven orders of magnitude by performing measurements with different exposure times, saturating detector pixels at the longer exposure times. Analysis of the stray light indicates about 4.4 % of the detected light is correctable stray light. An algorithm was then devised and implemented in the operational data processor to correct in-flight SWIR observations in near-real time, based on Van Cittert deconvolution. The stray light is approximated by a far-field kernel independent of position and wavelength and an additional kernel representing the main reflection. Applying this correction significantly reduces the stray-light signal, for example in a simulated dark forest scene close to bright clouds by a factor of about 10. Simulations indicate that this reduces the stray-light error sufficiently for accurate gas-column retrievals. In addition, the instrument contains five SWIR diode lasers that enable long-term, in-flight monitoring of the stray-light distribution.

## 1 Introduction

The Tropospheric Monitoring Instrument (TROPOMI) is the only instrument on board the ESA Copernicus Sentinel-5 Precursor satellite, which was launched on 13 October 2017 (Veefkind et al., 2012). The instrument maps the Earth atmosphere in two dimensions using two spectrometer modules, one covering the ultraviolet, visible and near-infrared (UVN) spectral ranges and the other covering the shortwave infrared (SWIR) spectral range 2305–2385 nm with a spectral resolution of 0.25 nm and a spectral sampling interval of 0.1 nm. The SWIR band is used for the retrieval of atmospheric CO and methane columns.

The spectral radiance from a ground swath of about 2600 km across track by about 7 km along track is imaged in consecutive periods of 1.08 s. The swath is partitioned into 216 ground pixels, each covering a viewing angle of $0.5°$. Hence, the spectrum from a given ground pixel can contain spatial stray light from the 215 other ground pixels, as well as spectral stray light from all ground pixels. To achieve the required accuracy of the spectral-radiance measurements, an accurate correction for the

stray light must be included in the data processing. The stray light has been characterized in detail with on-ground calibration measurements using a monochromatic quasi-point light source. Based on these measurements, an algorithm has been devised to correct the in-flight observations in near-real time.

The outline of the paper is as follows. The measurements are described in Sect. 2, followed by the first data processing in Sect. 3. The data are then examined in Sect. 4. An appropriate stray-light model and a correction algorithm are presented in Sects. 5 and 6, respectively. The further data processing to produce the input for the correction algorithm is explained in Sect. 7. Examples of corrected measurements are given in Sect. 8, followed by the conclusions in Sect. 9.

## 2   Calibration measurements

The SWIR and UVN spectrometers in TROPOMI share a common telescope. After the light is split into the different bands, it is imaged onto the SWIR spectrometer via relay optics and a second telescope. In the SWIR spectrometer (developed by SSTL, United Kingdom), the light passes an entrance slit etched in a metal coating on a wedge prism, a flat fold mirror, two collimator lenses, an immersed diffraction grating (produced by SRON), an anamorphic wedge prism, five imager lenses, a warm window, a cold detector window and finally hits the detector (Saturn model by Sofradir, France). The immersed grating consists of a silicon prism as the immersive medium with a diffraction grating on one surface. By illuminating the grating from inside the prism, the resolving power is increased by the refractive index of silicon. This allows the spectrometer to be much smaller than with a conventional echelle grating. More details of the spectrometer and specifically the immersed grating are given by Van Amerongen et al. (2017). The photovoltaic HgCdTe detector consists of a pixel array with 256 rows and 1000 columns, hybridized onto a read-out integrated circuit based on silicon complementary metal-oxide-semiconductor (CMOS) technology. In each detector pixel, the signal charge is converted into a voltage by a capacitive transimpedance amplifier (CTIA) and after a given exposure time stored in a sample-and-hold circuit before read-out. Details of the detector read-out and characterization are given by Hoogeveen et al. (2013). The spectral direction is imaged along the rows and the across-track spatial direction along the columns. The first collimator lens, the second imager lens and the warm window are made of germanium, while the other transmitting optics are made of silicon. The entrance slit has a bandpass filter with a reflectance of about 6 % in the operational range. The detector, the immersed grating and all other transmitting optical surfaces have antireflection coatings with a reflectance of 10 %, 0.3 % and 0.1 %, respectively.

The measurements for the SWIR stray-light calibration were performed at the Centre Spatial de Liège (CSL) in Belgium in February 2015, during the on-ground calibration campaign of TROPOMI (Kleipool et al., 2018). In these measurements, a quasi-point light source at a given across-track (swath) angle and with a given wavelength is imaged on the SWIR detector. The swath-angle range and the wavelength range are sampled independently, forming a large grid of measurements.

The setup is shown schematically in Fig. 1. The light source is a 2 W continuous-wave optical parametric oscillator (OPO), custom built by VSL (the Netherlands' national metrology institute). The OPO is pumped by a single-frequency distributed feedback (DFB) fiber laser operating at 1064 nm which is amplified to 10 W by an ytterbium fiber amplifier. The OPO wavelength is set coarsely between 2290 nm and 2390 nm by manually setting the temperature of the periodically poled lithium

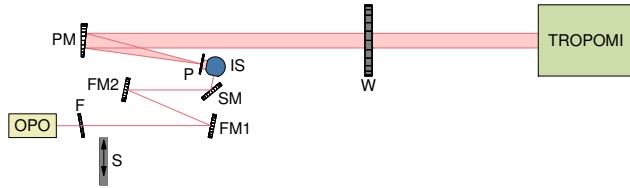

**Figure 1.** The setup for the stray-light measurements. The elements after the OPO are neutral density filter F, shutter S, fold mirrors FM1 and FM2, spinning mirror SM, integrating sphere IS, field stop P, parabolic mirror PM and window W of the vacuum chamber containing the TROPOMI instrument.

niobate (PPLN) crystal and rotating the etalon mounted on a galvo. The wavelength is scanned in steps of about 0.8 nm, with a repeating pattern of 1 manual step followed by 3 automatic steps, applying a changing piezo voltage to the fiber laser and simultaneously changing the crystal temperature with a predetermined dependence on the piezo voltage. To remove the speckle pattern from the light, it is sent to a 3.3 inch integrating sphere via a spinning mirror (1 inch diameter, 76 Hz) with an angle of
$1°$ between the rotation axis and the normal. The light exits the integrating sphere and is collimated with a field stop (9.6 mm diameter) and an off-axis parabolic mirror (500 mm focal length). The beam enters TROPOMI through the radiance port of the common telescope with a swath-angle coverage of $1.1°$. The instrument is mounted on a cradle in order to scan all swath angles in a range of $108°$ around nadir. Background measurements are taken by closing a shutter in front of the spinning mirror. Background data contain the same pixel-dependent offset, detector dark current and thermal background as light data taken at
the same exposure time.

At each wavelength, the swath angle is scanned in steps of $1.1°$, moving the light peak from top to bottom on the detector, followed by a scan in the opposite direction with the shutter closed. At each swath angle, detector images (frames) are taken at four different exposure times with about 20-fold increment steps (0.2, 4.6, 106 and 1998 ms), without changing any other setting. With a neutral density filter just after the OPO, the spectral radiance at the instrument is reduced to 200 times the
highest value in nominal operations. At this radiance, the detector is never saturated at the shortest exposure time. However, the signal with this exposure time is noisy away from the peak. At the three longer exposure times, the signal has a much better signal-to-noise ratio away from the peak, but the peak is saturated. These three exposure times are used to increase the dynamic range of the image by four orders of magnitude (see Sect. 3). At the three shorter exposure times at least 9 frames are averaged, but only 3 frames are taken at the longest exposure time in order to limit the total measurement period.
The wavelength was difficult to set accurately and was sometimes unstable during measurements. This caused some gaps, which were filled by extra scans performed after the main measurement series. In total, a slightly irregular grid was measured of 116 wavelengths by 99 swath angles. The peak positions are shown in Fig. 2, connected by lines per swath-angle scan and identically coloured for a manual scan and any following automatic scans.

A coarser grid with more frames per point would give the same signal-to-noise ratio in the final calibration data used to
correct measurement data, but this fine grid ensures that no important features only closer to specific wavelengths or swath angles are missed. The total measurement period was reduced to 113 hours by skipping many background measurements. At

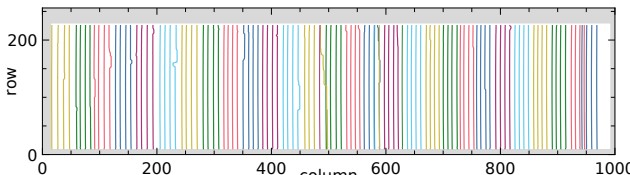

**Figure 2.** Fitted peak positions, connected by lines per swath-angle scan. Consecutive scans between manual steps have the same, arbitrary colour. In the grey area, the light is blocked by the entrance slit of the spectrometer (top and bottom) or a shield at the detector (left and right). Light scans were performed from top to bottom and (except 4 single scans after the main scans) from right to left.

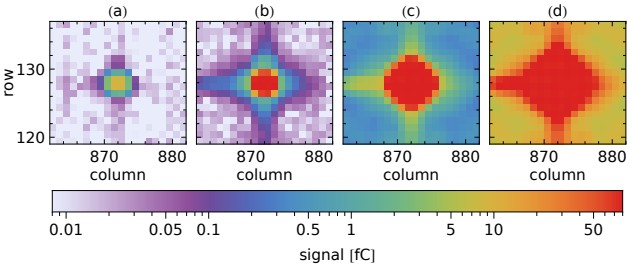

**Figure 3.** Background-corrected light peak at different exposure times: (a) 0.2 ms, (b) 4.6 ms, (c) 106 ms, (d) 1998 ms. Only at the shortest exposure time no pixels are saturated.

exposure time 4.6 ms, some of the grid points have no corresponding background measurements. Therefore the median of all background measurements at 4.6 ms is combined with each light measurement at 4.6 ms. At exposure times 0.2 ms and 106 ms, no background measurements were taken at all. In these cases, the median of the light measurements is used as background measurements at all swath angles and wavelengths. The peak in a given detector area only occurs in a small subset of the light measurements, which means it has a small effect on the median. In the area far away from the peak, the median may not constitute an accurate background, but for this area only data at the longest exposure time of 1998 ms will be used. In that case background measurements are available for each combination of swath angle and wavelength.

Both light and background measurements are offset-corrected, but the background measurements are not yet subtracted from the light measurements, because the total signal (due to external light, thermal background and detector dark current) is needed to merge the frames at the different exposure times.

## 3 Merging frames with different exposure times

The measurements show a small spot on the detector that moves as a function of swath angle and wavelength. At each position, data have been taken at four exposure times. Figure 3 shows an example, where only a small area around the peak is shown for one illumination at the four exposure times. Then for each pixel, the signal is taken from the longest exposure time that does not saturate (a saturated signal is defined as a signal larger than 90 % of the maximum possible signal). The background signal

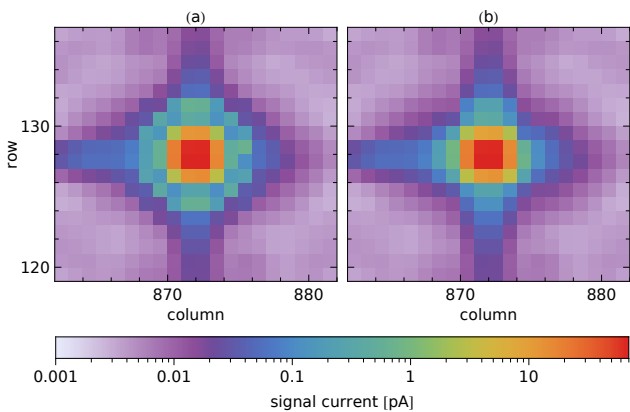

**Figure 4.** Combination of data in Fig. 3 (from signal to signal current) to increase the dynamic range, (a) without and (b) with blooming taken into account.

of this pixel at the chosen exposure time is subtracted, removing the pixel-dependent offset, detector dark current and thermal background. The result is divided by the exposure time to get a signal rate in digital counts per second. By applying a separate calibration of the number of electrons per count, it is converted to a signal current. By combining the data like this, the signal current can be measured with a dynamic range of more than seven orders of magnitude.

Without further measures, the result of merging the data taken with different exposure times would look like Fig. 4a. Due to the use of a CTIA in each detector pixel, the signal does not affect the detector bias voltage and signals of neighbouring pixels do not affect each other, unless a signal saturates. In that case blooming occurs, seen as a ring around the peak in Fig. 4a: the signal of unsaturated pixels becomes too high due to spilling from a direct neighbour pixel saturated by light, not dark current. Hence, for these pixels, the signal is taken from the exposure time that is one step shorter. The corrected result is shown in

Fig. 4b. The reason why blooming does not occur due to dark current is unknown and not investigated further as there are only about 80 detector pixels with a large enough dark current.

     Two corrections have been applied to the measured signals at an exposure time of 0.2 ms. First, a correction is needed for pixels with a large light signal that alternates between a lower and a higher value from frame to frame. This effect was not observed during the detector characterization in May–June 2013 (Hoogeveen et al., 2013) and is not understood. However,

a correction has been derived, based on frames with slightly smaller signals without alternating readings. Second, in the conversion to signal current, nominal exposure time 0.2 ms is actually replaced by the value 0.14 ms. This is based on stray-light correction of diode-laser, OPO and arc-lamp measurements, which produces consistent intermediate results. The two effects are probably caused by the very large signal currents in the light peak (200 times the highest value in nominal operations) and an exposure time at the limit of the detector specification.

After these steps, each merged frame is corrected for pixel response nonuniformity (PRNU). This is the pixel-to-pixel variation of the gain (ratio of the signal current to the input photon rate), determined in separate calibration measurements using a light source with a flat spectrum. This correction is very small, up to about 1 %.

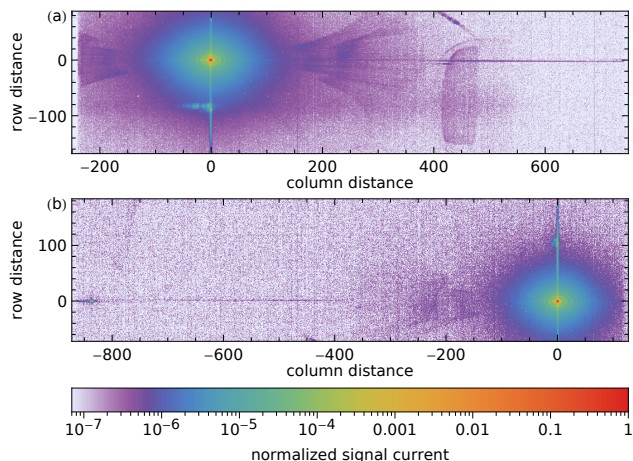

**Figure 5.** Merged frames at two combinations of wavelength and swath angle, normalized to a peak value of 1: (a) 2320.20 nm and $-23.8°$, (b) 2377.90 nm and $+23.5°$.

At this stage, a merged frame consists of signal current $I_{\text{meas}}[r,c]$ as a function of detector row $r$ and column $c$. The area near the peak is fitted with a two-dimensional function $S(r,c)$. The one-dimensional convolution of a Gaussian distribution with standard deviation $\sigma$ and a uniform distribution with mean 0 and full width $w$ is given by

$$\mathcal{B}(x;\sigma,w) = \frac{1}{2w}\left[\text{erf}\left(\frac{x+w/2}{\sqrt{2}\sigma}\right) - \text{erf}\left(\frac{x-w/2}{\sqrt{2}\sigma}\right)\right]. \tag{1}$$

The fit function is

$$S(r,c) = a\,\mathcal{B}(r-r_0;\sigma_{\text{spat}},w_{\text{spat}})\,\mathcal{B}(c-c_0;\sigma_{\text{spec}},w_{\text{spec}}), \tag{2}$$

with spatial (vertical) peak position $r_0$, spectral (horizontal) peak position $c_0$, integrated signal current $a$, spatial width parameters $\sigma_{\text{spat}}$ and $w_{\text{spat}}$ and spectral width parameters $\sigma_{\text{spec}}$ and $w_{\text{spec}}$. The variation of the light intensity between frames is removed by normalizing the frame by the integrated signal current:

$$I[r,c] = I_{\text{meas}}[r,c]/a. \tag{3}$$

The values of the four width parameters are not used, only the fitted peak position $(r_0, c_0)$ is needed later on. In 10 % of the frames, either the peak is too close to the detector edges or one of the fitted widths is too small. After discarding these cases, there are still 10361 valid frames left.

## 4   Examination of data

Figure 5 shows two examples of merged frames, at 2320 nm and 2378 nm and at opposite swath angles. The electronic noise has been suppressed to about $10^{-7}$ times the peak signal. It determines the lower end of the dynamic range. The halo around

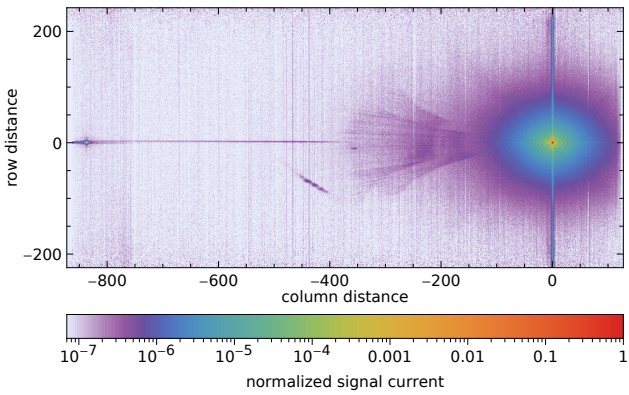

**Figure 6.** At one wavelength (2377.90 nm), the median frame over all swath angles, normalized to a peak value of 1.

the peak is light scattered from optical surfaces, possibly due to polish imperfections with low spatial frequencies. The vertical line through the peak is pure spatial stray light from optics before the spectrometer slit: the common telescope and the relay optics. The line is slightly curved due to the variation of about 0.5 column at a given wavelength but different swath angles, known as the spectral smile. The weak horizontal line through the peak is due to imperfections of the grating line positions. It

is slightly rotated, as can be seen at the right side of Fig. 5a. The origin of the wings 150 to 400 columns from the peak on both sides is unknown. See video S1 in the Supplement for two sequences of frames, at 23.5° and at 2320.20 nm.

The unfocussed spot straight below the peak (Fig. 5a) or above it (Fig. 5b) is due to a double reflection, first at the detector, then at the front side of the immersed grating, from the inside. This means the light is diffracted three times in the same grating order. According to the optical model, below 2360 nm the reflected beam hits the edge of the immersed grating surface, causing

the measured feature to stretch horizontally (Fig. 5a). The reflected beam covers the margin of about 1.5 mm at the surface edge without antireflection coating, which makes this the strongest feature moving relative to the peak.

The features described above will be reduced by the correction algorithm. Other features depend too much on wavelength or swath angle and remain after the correction, but they have values six to seven orders lower than the peak value. One is seen in Fig. 5a as a large block shape in the column range 400–500. It is due to a double reflection, first at the detector, then at the

15 front of the fourth imager lens.

To examine the data more closely, all frames at a given wavelength are shifted vertically until the peaks overlap within a pixel. The median for each pixel at 2377.90 nm is shown in Fig. 6. The reflection 110 pixels above the peak in Fig. 5b has disappeared in the median image, because it moves vertically relative to the peak position. Note also the spot 836 pixels to the left of the peak in Fig. 6. It remains at the same vertical position as the peak, but moves horizontally with wavelength. It is

20 caused by light that diffracts twice in the same order in the immersed grating, with a reflection at the front surface in between.

For an examination of the central area, the scan data are combined differently. For Fig. 6, the original frames were shifted vertically by an integer number of pixels. In Fig. 7, the shift is over a non-integer number to bin the result at subpixel resolution, before the median is taken per bin. This shows the structures near the peak more clearly. The data from two swath-angle scans

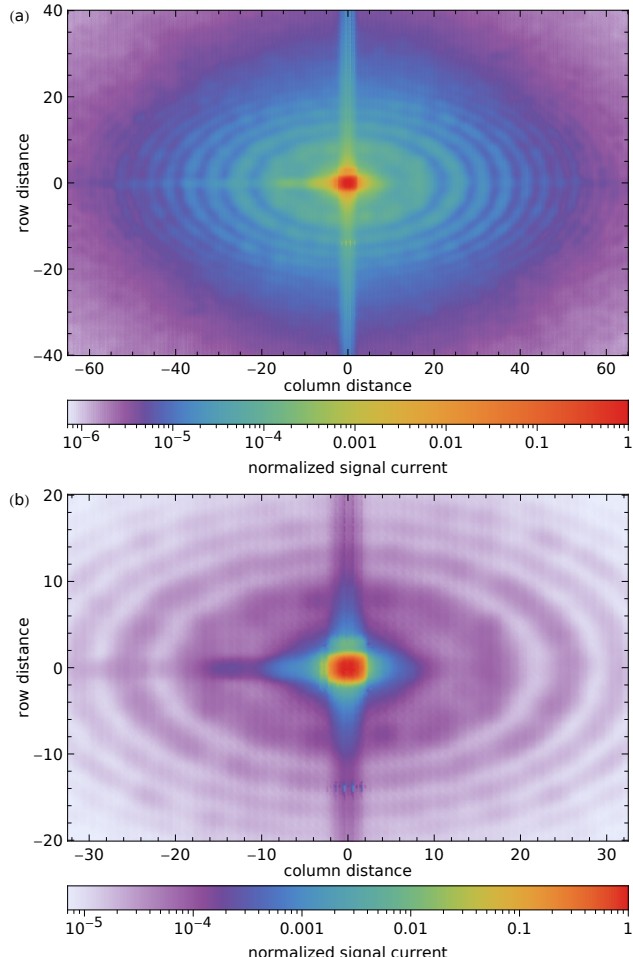

**Figure 7.** Two views of the peak area at a resolution of 0.2 pixel, binning data at 2319.55 nm and 2320.20 nm.

within a wavelength range of 0.65 nm are combined with bins of 0.2 pixel. All bins are filled, due partly to the spectral smile. The rings in Fig. 7 have slightly decreasing diameters with increasing wavelength. Their spacing is consistent with interference of light from two separated surfaces. The radially decreasing brightness suggests interference of scatter, not reflections. The elliptical shape indicates that they are probably generated before the grating surface. The remaining surfaces within the spectrometer are unlikely candidates. At the entrance pupil before the spectrometer, the surface separation needs to be 6 mm vacuum to explain the fringe diameters. The fringes cannot originate in the light source, because they are also seen using the on-board diode lasers. The purple rings in Fig. 7b look like bead strings and the gaps between the beads in different rings form hyperbolas. Their origin is not understood, but it may be related to the origin of the ring structures.

Figure 8 shows a vertical cross section through the peak on a linear and logarithmic scale (after binning the shifted original frames over intervals of 0.025 pixel), including a fit with the convolution of a Gaussian and a uniform distribution. The

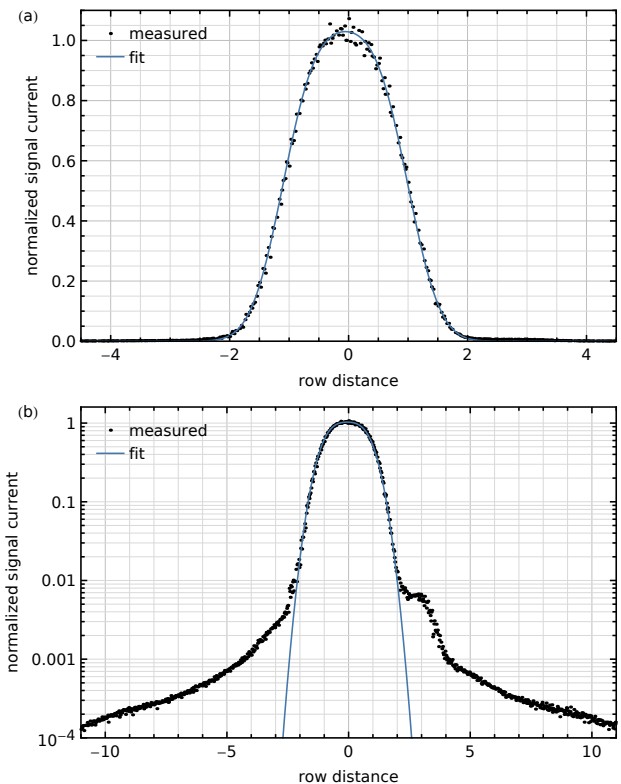

**Figure 8.** Vertical (spatial) cross section through the peak using the combined data at 2377.90 nm on (a) a linear scale and (b) a logarithmic scale. Included is a fit with the convolution of a Gaussian and a uniform distribution.

maximum is not exactly 1, because the same normalization factor has been used as in Fig. 6 despite different binning. The full width at half maximum of 2.1 spatial pixels corresponds to the stimulus size of about $1.1°$, in agreement with the instrument design model. The signal 3 pixels above the line or peak is 0.5 % higher than the 0.2 % expected from the neighbouring signals. This bump is probably a feature of the stimulus and not the instrument, because it was not seen in measurements with a xenon
5   arc lamp.

Figure 9 shows a horizontal cross section using bins of 0.05 pixel and the average instrument spectral response function (ISRF) at the peak positions used. The ISRF was determined with different, dedicated measurements (Van Hees et al., 2018). Here it is only scaled vertically to fit the data. It is defined over the central 9 pixels that is used in trace-gas retrievals, but in the logarithmic plot the ISRF is extrapolated beyond that range. The 0.1 % bump centred 7 pixels to the left of the peak seems to
10   be extra stray light, not a laser feature, because it has been seen with the on-board diode lasers as well.

The vertical and horizontal cross sections contain more stray light than any other cross section through the peak. Figure 10 shows the vertical and horizontal cross sections together with a diagonal cross section, taken at $45°$. In all three cases, a distance

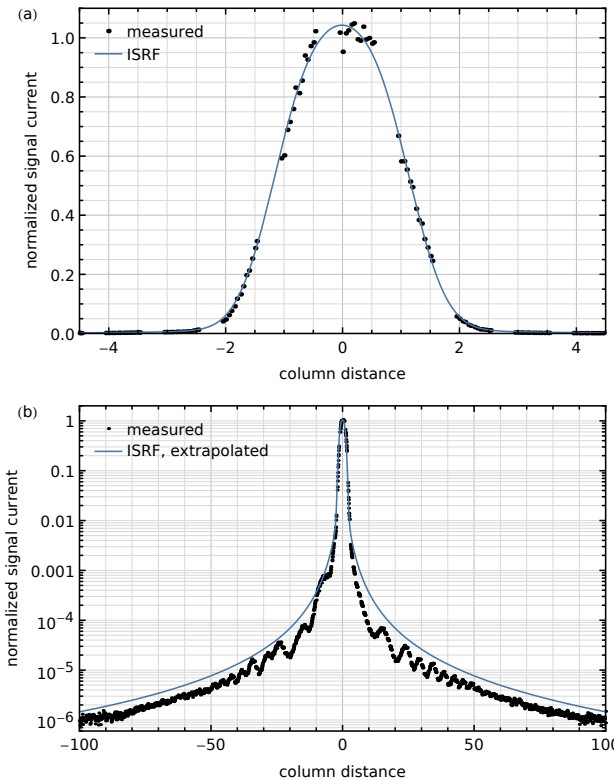

**Figure 9.** Horizontal (spectral) cross section through the peak using the combined data at 2377.90 nm on (a) a linear scale and (b) a logarithmic scale. Included is the corresponding ISRF with a fitted height, extended at absolute column distances larger than 4.5.

of 1 pixel corresponds to 30 μm on the detector. The reflection that moves vertically relative to the peak position is not fully removed by using a median per data bin, causing the feature at the lower end of the vertical cross section.

## 5 Stray-light model

Once the general behaviour of the stray light was examined, a descriptive model was defined, starting with some terminology. A
5 "spread function" maps an object to image space, which involves many detector pixels. A "response function" maps an image to object space, which is a property of a given detector pixel. A qualifier describes the dimensions involved in the mapping: the point response function (PRF) is used for the two spatial dimensions (across-track and along-track), the ISRF for the spectral dimension and the spatial-spectral spread function (SSSF) for the combination of the across-track spatial dimension and the spectral dimension. While the PRF and ISRF are seen as only descriptive, the part of the SSSF away from the centre is seen
10 as stray light that will be corrected. Stray light is only corrected when the source itself is imaged on the detector, because the direct source light serves as input for the correction.

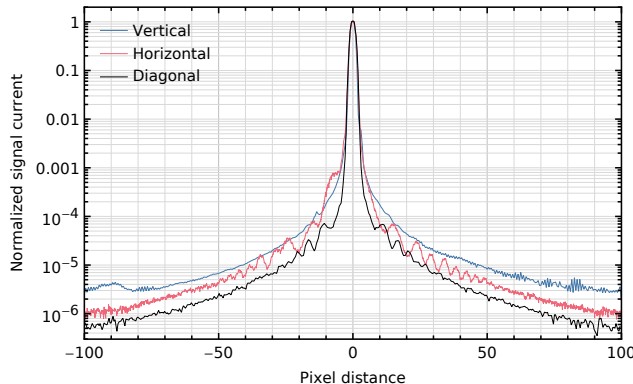

**Figure 10.** Vertical (spatial), horizontal (spectral) and diagonal cross sections through the peak on a logarithmic scale, using the combined data at 2377.90 nm.

Any system will have an SSSF contribution that is independent of the peak position as well as contributions that are specific for a peak position. The latter are usually more complicated to correct for. Fortunately, it turns out that these are small in the TROPOMI-SWIR system. Hence, the model for the SSSF as a function of swath angle and wavelength contains only the following two contributions:

5 – The median of images of a monochromatic quasi-point source over all swath angles and wavelengths, after they have been shifted to let the peaks coincide, called the "stable kernel", $\mathbf{K}_{\text{stable}}$. This kernel has odd dimensions, has the median peak at the centre and is normalized so the sum of all elements is 1. It is split into a near-field kernel $\mathbf{K}_{\text{near}}$ and a far-field kernel $\mathbf{K}_{\text{far}}$, by using a mask $\mathbf{M}_{\text{far}}$ with values in the range $[0,1]$:

$$\mathbf{K}_{\text{far}} = \mathbf{M}_{\text{far}} \circ \mathbf{K}_{\text{stable}}, \tag{4}$$

10 $$\mathbf{K}_{\text{near}} = \mathbf{K}_{\text{stable}} - \mathbf{K}_{\text{far}}, \tag{5}$$

where $\circ$ is the element-wise product. A centred area of 7 spatial by 9 spectral elements in $\mathbf{M}_{\text{far}}$ is set to 0 and the rest to 1.

– A dim, blurry version of the peak near the same column that moves downward when the peak is moved upward. This main reflection is described by the median of all images, after the stable kernel has been subtracted and the images have 15 been shifted to let the reflections coincide. This is the "reflection kernel", $\mathbf{K}_{\text{refl}}$. The main reflection is not already part of the stable kernel, because it moves too much to be noticed in a median. This kernel has also odd dimensions and is normalized so the sum of all elements is 1, but neither the reflection nor the (subtracted) peak is at the centre (see Appendix A). Intensity variations of the reflection are accounted for by detector map $\mathbf{E}_{\text{refl}}$, where each pixel is set to the relative intensity of the reflection when the main peak is at that pixel.

Contributions of other features moving relative to the peak are neglected. Stray light, i.e. the SSSF part that is corrected, consists of $\mathbf{K}_{\text{far}}$ and $\mathbf{K}_{\text{refl}}$. The light in $\mathbf{K}_{\text{near}}$ is considered as the ISRF in the spectral direction and one dimension of the PRF in the spatial direction.

## 6   Correction algorithm

If $\mathbf{F}$ represents an ideal frame without stray light, the stable part of the stray light is given by $\mathbf{K}_{\text{far}} \otimes \mathbf{F}$, where $\otimes$ indicates a convolution. The integrated signal in the stray light is a fraction $\sum_{k,l}(\mathbf{K}_{\text{far}})_{k,l}$ of the total integrated signal. Measured frame $\mathbf{J}_0$ is the sum of the ideal frame and the stray light, taking into account that the stray light is removed from the ideal frame:

$$\mathbf{J}_0 = \left(1 - \sum_{k,l}(\mathbf{K}_{\text{far}})_{k,l}\right)\mathbf{F} + \mathbf{K}_{\text{far}} \otimes \mathbf{F}. \tag{6}$$

The stable part of the stray light is corrected with Van Cittert deconvolution (Van Cittert, 1931; Berry and Burnell, 2000), one of the simplest methods of iterative image restoration: given an input frame $\mathbf{J}_0$, the frame after $i$ iterations is

$$\mathbf{J}_i = \frac{\mathbf{J}_0 - \mathbf{K}_{\text{far}} \otimes \mathbf{J}_{i-1}}{1 - \sum_{k,l}(\mathbf{K}_{\text{far}})_{k,l}}. \tag{7}$$

The number of iterations has been set to $n = 3$. Trials have shown that a higher number does not improve the correction, while adding load to the processor. The convolution is performed with zero-padding outside the frame and the result has the same dimensions as the frame. The rationale of this algorithm is as follows: term $\mathbf{K}_{\text{far}} \otimes \mathbf{J}_{i-1}$ is an approximation of the stray light and is subtracted from original frame $\mathbf{J}_0$. The result is closer to the perfect image, and hence also a better input to derive an approximation of the stray light, which is determined in the next iteration. The denominator constitutes a normalization. This is made clearer by writing the first iteration as

$$\mathbf{J}_1 = \frac{\mathbf{J}_0 - \mathbf{K}_{\text{far}} \otimes \mathbf{J}_0}{1 - \sum_{k,l}(\mathbf{K}_{\text{far}})_{k,l}} = \mathbf{K}_1 \otimes \mathbf{J}_0, \tag{8}$$

with

$$\mathbf{K}_1 = \frac{\mathbf{\Delta} - \mathbf{K}_{\text{far}}}{1 - \sum_{k,l}(\mathbf{K}_{\text{far}})_{k,l}}, \tag{9}$$

where $\mathbf{\Delta}$ is a two-dimensional Kronecker delta, i.e. a matrix with one non-zero element 1 at the centre. The iteration consists of a convolution with a kernel $\mathbf{K}_1$, for which the sum of all elements is 1. Hence, the stray light is not removed but redistributed.

Frame $\mathbf{J}_n$ after the last iteration in the correction of the stable part of the stray light, using Eq. (7), is then corrected for the reflection part:

$$\mathbf{J}_{\text{corr}} = \mathbf{J}_n - \mathbf{K}_{\text{refl}} \otimes (\mathbf{E}_{\text{refl}} \circ \mathbf{J}_n)^{\text{R}}, \tag{10}$$

where R indicates the matrix operation of reversing the row order. The reflection kernel has been defined in such a way that this reversing operation maps the main reflection at the correct position, as explained in Appendix A.

## 7 Determination of the calibration data

For the correction, three types of calibration data need to be determined: stable kernel $\mathbf{K}_{\text{stable}}$, reflection kernel $\mathbf{K}_{\text{refl}}$ and relative reflection intensity $\mathbf{E}_{\text{refl}}$. First the stable kernel is calculated. A merged frame at a given swath angle $\alpha$ and wavelength $\lambda$ contains signal current $I[r,c,\alpha,\lambda]$ as a function of detector row $r$ and column $c$. The fitted peak position is $(r_{\alpha,\lambda}, c_{\alpha,\lambda})$. Using linear interpolation, an extended array is produced where the peak is at the centre:

$$K[y,x,\alpha,\lambda] = I\big[y+r_{\alpha,\lambda}, x+c_{\alpha,\lambda}, \alpha, \lambda\big], \tag{11}$$

with integers $y \in [-255, +255]$ and $x \in [-999, +999]$. At coordinates outside the detector range, $I[r,c,\alpha,\lambda]$ is assumed to be a dummy value. Note that the resulting array is twice as wide and high as a detector frame. The median of the set of arrays $K[y,x,\alpha,\lambda]$ over all $\alpha$ and $\lambda$ is $K[y,x]$. In this calculation, dummy values are discarded and if there are no valid values left for a given element, it is set to zero. Afterwards, edge columns and rows that only contain zeros are removed where possible, as long as element $K[0,0]$ with the peak can be kept at the centre of the array. Stable kernel $\mathbf{K}_{\text{stable}}$ is $K[y,x]$ after normalization so the sum of all elements is 1. The amount of stray light described by the stable kernel can now be determined after applying Eq. (4): it is a fraction $\sum_{k,l}(\mathbf{K}_{\text{far}})_{k,l} = 0.043$ of the detected light.

The stable kernel is shown in Fig. 11 in three ranges. Figure 11c can be compared with Fig. 7a: they show the same area with the same colour scale, but Fig. 7a uses basically one wavelength and has a higher resolution. Because the ring sizes depend slightly on wavelength, the larger rings are averaged away in the stable kernel. The spot in Fig. 6, 836 pixels to the left of the peak, moves slowly relative to the peak as a function of wavelength, and has become stretched and weakened in the stable kernel.

For the determination of the reflection kernel, only merged frames $I[r,c,\alpha,\lambda]$ are used where the main reflection is separated enough from the peak to be determined accurately, $r_{\alpha,\lambda} \notin [82, 173]$. The stable kernel is shifted and subtracted from each frame, leaving the normalized rest of the stray light:

$$I_{\text{rest}}[r,c,\alpha,\lambda] = I[r,c,\alpha,\lambda] - K_{\text{stable}}\big[r - r_{\alpha,\lambda}, c - c_{\alpha,\lambda}\big], \tag{12}$$

using linear interpolation in $K_{\text{stable}}$. The results are shifted by $\Delta r$ rows and $\Delta c$ columns until the main reflection is found in an area centred at position $(r_{\text{centre}}, c_{\text{centre}})$, with $r_{\text{centre}} = 127.5$ and arbitrary $c_{\text{centre}}$. This is performed by interpolating values $I_{\text{rest}}[r - \Delta r, c - \Delta c, \alpha, \lambda]$, where $\Delta r = r_{\alpha,\lambda} - r_{\text{centre}}$ and $\Delta c = c_{\text{centre}} - c_{\alpha,\lambda}$ (see Appendix A). Expressed in coordinates $y = r - r_{\text{centre}}$ and $x = c - c_{\text{centre}}$, the new array is given by

$$I_{\text{refl}}[y,x,\alpha,\lambda] = I_{\text{rest}}\big[y + 255 - r_{\alpha,\lambda}, x + c_{\alpha,\lambda}, \alpha, \lambda\big], \tag{13}$$

with integers $y \in [-78, +78]$ and $x \in [-49, +49]$. The reflection kernel and relative reflection intensity $E_i[\alpha,\lambda]$ are then determined iteratively. Iteration $i = 1$ starts with $E_0[\alpha,\lambda] = 1$. The median of the set of arrays $I_{\text{refl}}[y,x,\alpha,\lambda]/E_{i-1}[\alpha,\lambda]$ over all $\alpha$ and $\lambda$ is $k_i[y,x]$. Elements $k_i[y,x] < 0.01\max_{y,x}(k_i[y,x])$, which contain essentially noise, are set to zero. The result is normalized so the sum of all elements is 1, producing preliminary reflection kernel $K_{\text{refl},i}[y,x]$. The corresponding relative

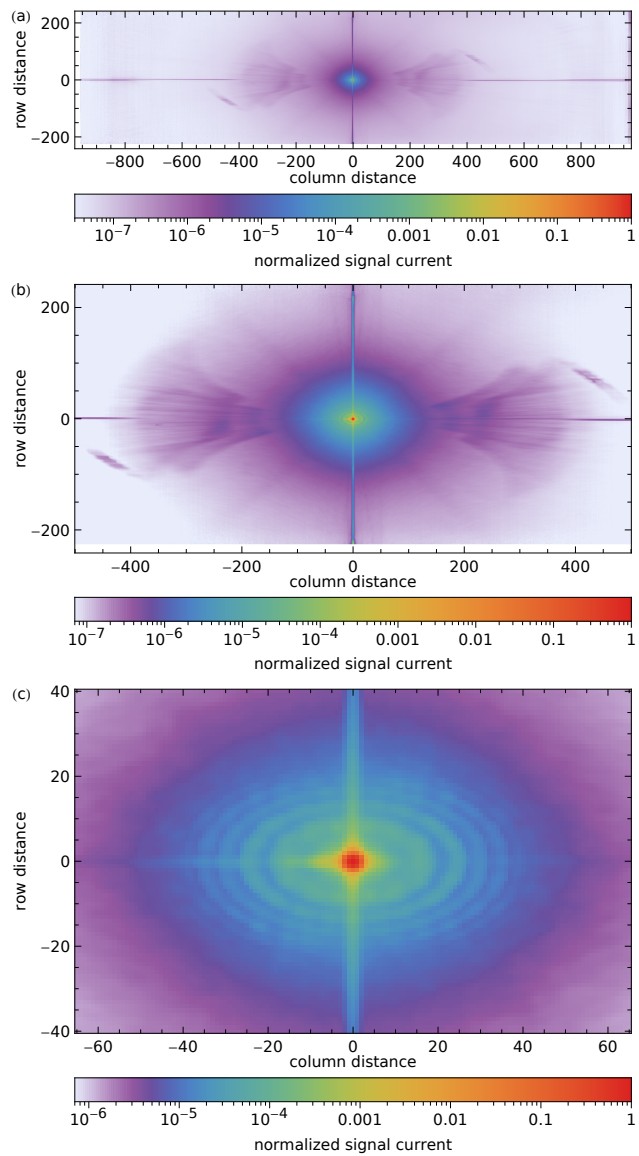

**Figure 11.** The stable kernel in three ranges: (a) full range, (b) main part, (c) same range as Fig. 7a. For easy comparison, the data have been multiplied by 6.44 in order to get a peak value of 1. White edges have been set to 0.

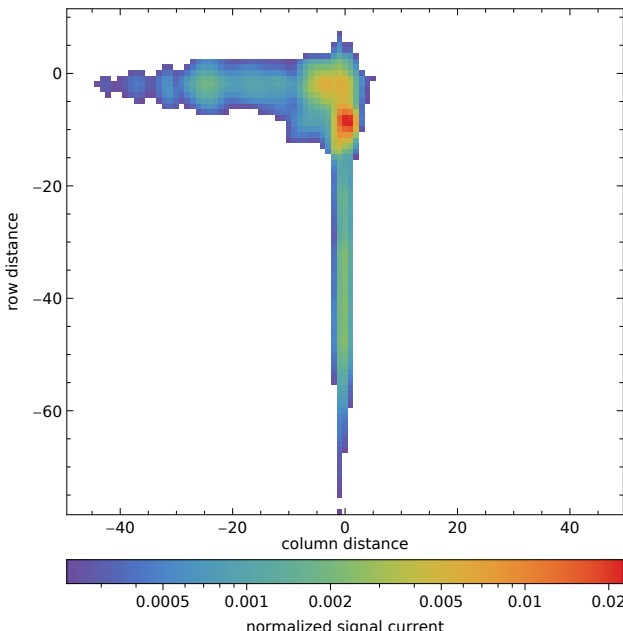

**Figure 12.** The lower part of the reflection kernel. White pixels have been set to zero as they contain essentially noise. The upper part, making the area symmetric around the origin, contains only zeros.

reflection intensity is determined by comparing the remaining measured stray light with a scaled reflection kernel, where the scaling is optimized:

$$E_i[\alpha, \lambda] = \arg\min_{\epsilon} \sum_{y,x} \left( I_{\text{refl}}[y, x, \alpha, \lambda] - \epsilon K_{\text{refl},i}[y, x] \right)^2. \tag{14}$$

Only two iterations are needed for convergence: the final reflection kernel is $\mathbf{K}_{\text{refl}} = K_{\text{refl},2}[y, x]$, shown in Fig. 12. Relative reflection intensity $E_2[\alpha, \lambda]$ is given in Fig. 13a. It is a function of peak parameters $r_{\alpha,\lambda}$ and $c_{\alpha,\lambda}$ and needs to be converted to a scaling map $\mathbf{E}_{\text{refl}}$ as a function of detector pixel. This is performed by fitting a third-order bivariate polynomial (Figs. 13b and 13c), which also reduces noise and fills gaps. The model is

$$
\begin{aligned}
E_{\text{fit}}(y, x; \boldsymbol{a}) = a_0 \\
+ a_1 y + a_2 x \\
+ a_3 T_2(y) + a_4 xy + a_5 T_2(x) \\
+ a_6 T_3(y) + a_7 x T_2(y) + a_8 y T_2(x) + a_9 T_3(x),
\end{aligned} \tag{15}
$$

with scaled indices

$$
\begin{aligned}
y = 2(r_{\alpha,\lambda}/255) - 1, \\
x = 2(c_{\alpha,\lambda}/999) - 1
\end{aligned} \tag{16}
$$

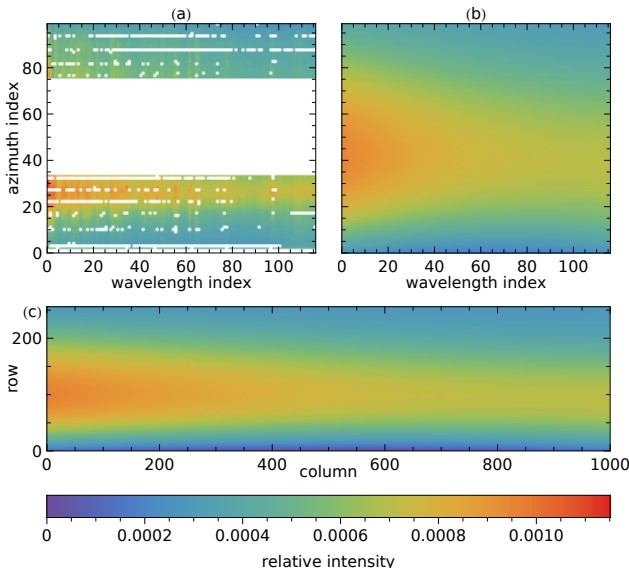

**Figure 13.** Maps of the relative intensity of the reflection kernel: (a) measured values, sorted by peak wavelength and swath angle, with unused values shown white; (b) fitted values on the same grid; (c) fitted values interpolated as a detector map.

and Chebyshev polynomials of the first kind $T_i(z)$. The model values for $r_{\alpha,\lambda} \in [0, 255]$ and $c_{\alpha,\lambda} \in [0, 999]$ form map $E_{\text{refl}}[r_{\alpha,\lambda}, c_{\alpha,\lambda}]$. The stray-light fraction in the detected light described by $\mathbf{K}_{\text{refl}}$ is $(\mathbf{E}_{\text{refl}})_{k,l}$, which varies between $0.2 \times 10^{-3}$ and $1 \times 10^{-3}$ (Fig. 13).

## 8 Correction results

In this section, the result of the stray-light correction is shown for several cases. First, the correction of a merged frame is checked. Figure 14a shows the signal before and after correction. The vertical line through the peak is weaker after correction, but it is still visible because the deconvolution cannot take into account the spectral smile of 0.5 pixel. Also remnants of the ring structure can still be seen as the rings are slightly wavelength dependent. Most, but not all of the main reflection is corrected. The large feature at column 700 moves with respect to the peak and is thus not corrected.

Other measurements show the cumulative stray light when one detector dimension is illuminated: for spectral stray light a measurement is used with monochromatic light from one of the five on-board DFB diode lasers (Nanoplus, Germany) via a diffuser (Fig. 14b) and for spatial stray light a measurement with quasi-white light from an external xenon arc lamp (Fig. 14c). The diode lasers have wavelengths in the SWIR spectral range at roughly equal intervals and are used for in-flight monitoring of the ISRF and stray light. The measurements for Fig. 14b and c were performed with lower light intensities, leading to a smaller signal-to-noise ratio than available in the stray-light characterization. In these cases with illumination over many pixels and at regular intensities, the stray-light correction works well. Some stray light remains around the corrected diode-laser signal,

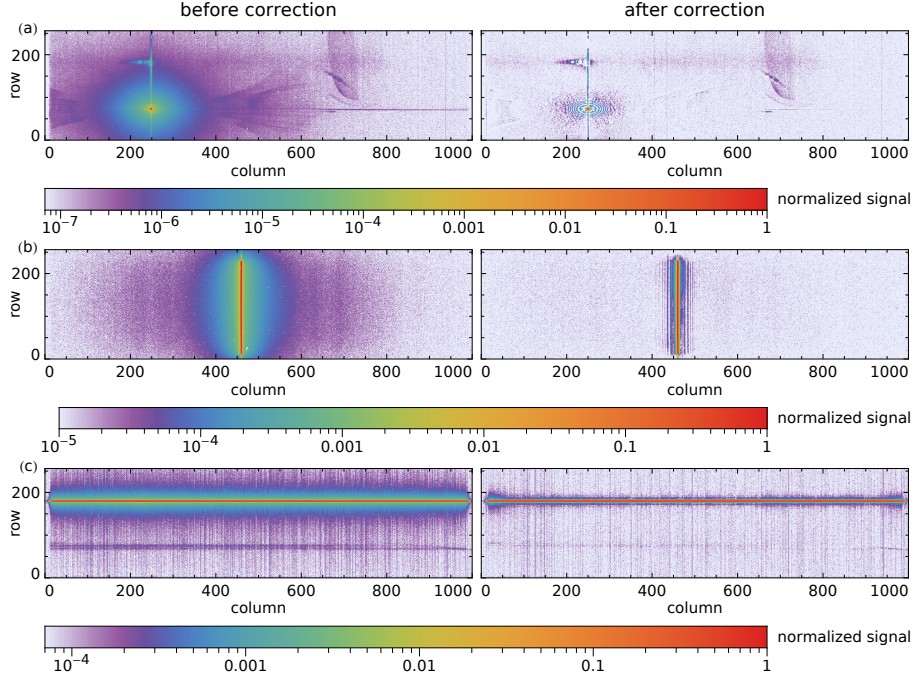

**Figure 14.** Examples of measurements before (left) and after (right) stray-light correction with different illumination patterns: (a) a monochromatic single swath-angle spot in one of the stray-light measurements; (b) a monochromatic full-swath line from an on-board diode laser; (c) a white-light single swath-angle line from an external xenon arc lamp. In each case, the maximum signal has been set to 1. Because the measured signal range varies between the examples, the minimum value of the colour scale is also different.

mainly due to the slight wavelength dependence of the rings around the peak. The horizontal line near row 70 in Fig. 14c is the sum of the main reflection over all wavelengths. It is a double line due to the irregular shape of this reflection, but most of it can be corrected. The extra stray light around the main line near the outer columns is attributed to light reflected from the detector shield limiting the wavelength range. Cross sections perpendicular to the lines in Figs. 14b and 14c are shown in Fig. 15.

5    To show the effect of the stray-light correction applied to an Earth radiance measurement, a swath is simulated which is half clear-sky forest (albedo 0.05) and half clouds (at 2 km height, albedo 0.40), assuming a U.S. Standard atmosphere, a solar zenith angle of $40°$ and a viewing zenith angle of $0°$ over the entire swath. The two radiance spectra are given in Fig. 16. The expected signal (without stray light), including the radiance responsivity and the spectral smile, is shown in Fig. 17a.

    Stray light is introduced by convolving the expected signal with the SSSF. However, if the stable kernel would be used as
10  an SSSF approximation for the whole detector, the effect of neglected features moving relative to the peak cannot be assessed. Instead, the detector is divided into areas of pixels around all peak positions found in the merged frames. A given measured merged frame constitutes the most accurate SSSF available for the detector pixels in the corresponding area. Care has to be taken with signal scaling and padding to the right dimensions before a merged frame can be used in a convolution. A merged frame is converted to a kernel by applying the same normalization factor as in the creation of the stable kernel and padding to

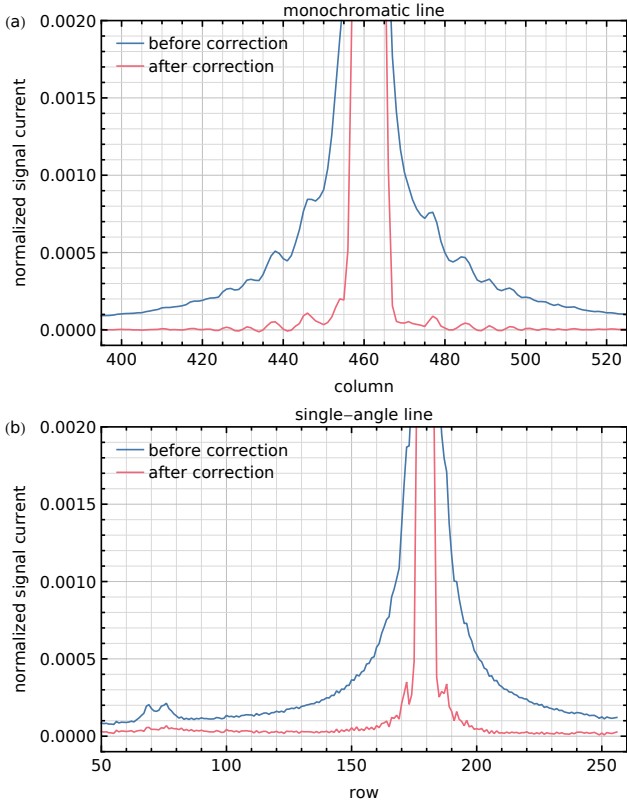

**Figure 15.** (a) Median horizontal cross section of the monochromatic line shown in Fig. 14b. (b) Median vertical cross section of the single swath-angle line shown in Fig. 14c. In each case, the maximum signal has been set to 1.

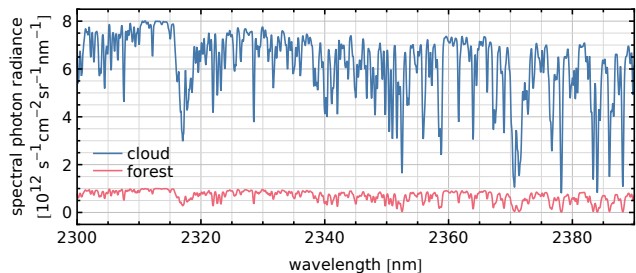

**Figure 16.** Spectra of cloud and forest scenes as used in the simulated Earth radiance measurements.

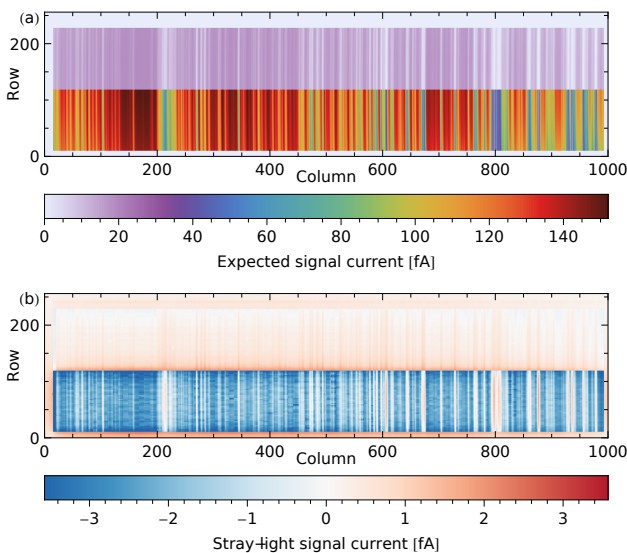

**Figure 17.** Simulation of an Earth radiance measurement: (a) the expected signal current without stray light, (b) the corresponding stray light.

the same dimensions with the peak at the centre. The padding values are the values in the stable kernel at the same positions. This is followed by multiplication with $\mathbf{M}_{far}$. The resulting kernel is used as $\mathbf{K}_{far}$ in Eq. (6) and applied to the corresponding area in the simulated radiance frame. This is repeated for all merged frames and corresponding areas in the radiance frame and the sum is taken.

The resulting signal with stray light looks very similar to the signal without in Fig. 17a, hence the difference is given in Fig. 17b. Note that stray light is not simply added light, but a redistribution of light. Effectively, some light is transferred from the bright cloudy scenes to the dark clear-sky scenes and the absorption lines. The data are somewhat grainy, because there is only one merged frame per area of about $3 \times 8$ detector pixels.

Figure 18 shows the stray light at a given pixel relative to the expected signal at the same detector position, before (Fig. 18a)
and after (Fig. 18b) correction. The stray light is relatively high where the expected signal is lowest: in the strongest water lines of the forest spectrum with an absorption by the atmosphere of 99.0 %. In the strongest line, the stray light is 430 % before correction and 30 % after correction, but these numbers will be twice as high in a more humid scene with an absorption of 99.5 %. The stray-light values depend less on the scene and wavelength when the stray light is considered as an absolute signal contribution instead of a relative one. It is expressed as a percentage of the maximum expected value on the same detector row,
corresponding to the continuum around 2313 nm (column 167) in the same scene. This is also in line with operational methane and CO retrieval, where the absolute difference between measurement and model is minimized (Hu et al., 2016; Landgraf et al., 2016).

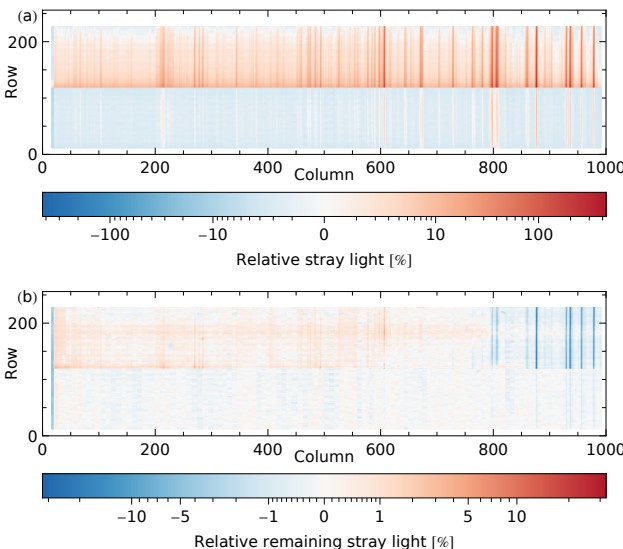

**Figure 18.** Simulation of an Earth radiance measurement: (a) the stray light expressed as a percentage of the expected signal at the same detector position, (b) the remaining stray light after stray-light correction expressed as a percentage of the expected signal at the same detector position.

The stray light normalized to the continuum of the scene is shown in Fig. 19a, with a maximum of about 10 % in the forest spectra that are closest to the cloud spectra. The remaining stray light after correction is given in Fig. 19b, with a maximum of about 1 % in the performance spectral range 2305–2385 nm (columns 74–944). The stray light in the forest spectrum located four rows from the cloud spectra before and after correction is given in Fig. 20 (relative to the expected signal current at the same detector position) and Fig. 21 (normalized to the expected continuum signal current).

By replacing fixed areas in each merged frame with the corresponding values of the stable kernel, the origin of some remaining features can be found. The positive band centred at row 180 in Fig. 19b is mostly due to shape variations of the main reflection. The rest of the remaining stray light in the upper half of Fig. 19b, both positive and negative, is mostly due to differences smaller than $5 \times 10^{-8}$ between the merged frames and the stable kernel, for example the low-signal features moving relative to the peak.

The correction reduces not only the general level of stray light, but also the sharp features near absorption lines (Fig. 21). This reduces the impact of the remaining stray light on retrieval. Applying non-scattering retrieval (without aerosol or cirrus parameters) to the forest spectrum used for Fig. 21, the error of the methane column reduces from 14 % before correction to 0.26 % after correction, within the error budget of 0.35 % for stray light. The error of the CO column reduces from 26 % before correction to 1.3 % after correction, within the stray-light error budget for the given CO column of 3.0 %. The forest spectra at other rows give similar results after correction.

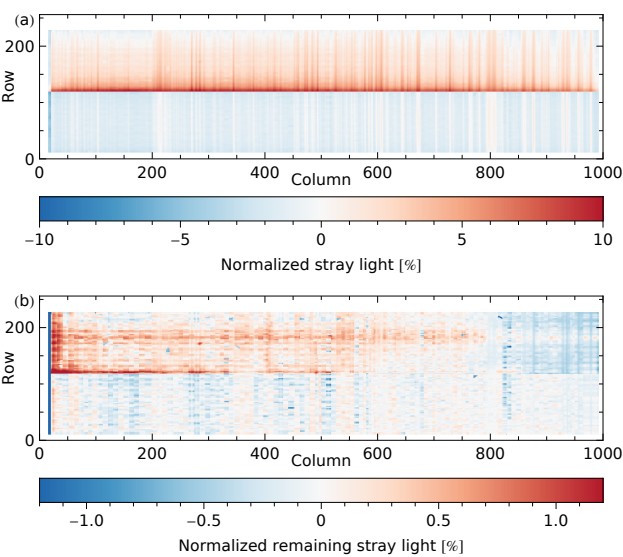

**Figure 19.** Simulation of an Earth radiance measurement: (a) the stray light expressed as a percentage of the expected continuum in the given row, (b) the remaining stray light after stray-light correction expressed as a percentage of the continuum.

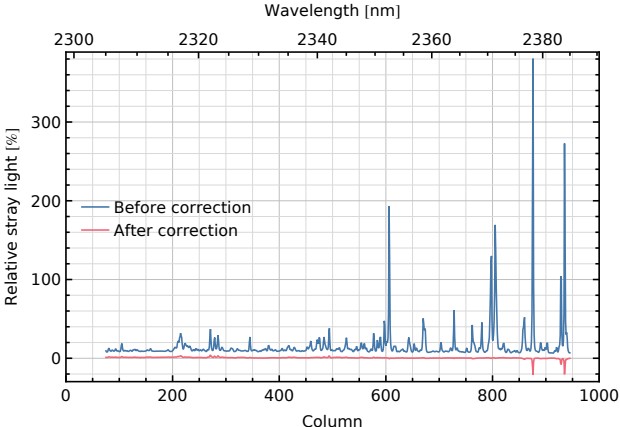

**Figure 20.** Stray light as a percentage of the expected signal current at the same detector position, before and after correction. These are cross sections of Figs. 18a and 18b, respectively, for the forest spectrum in performance range 2305–2385 nm located four rows from the cloud spectra.

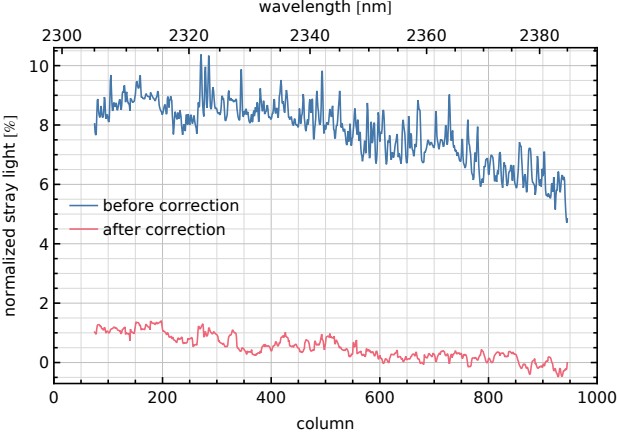

**Figure 21.** Stray light as a percentage of the expected continuum signal current, before and after correction. These are cross sections of Figs. 19a and 19b, respectively, for the forest spectrum in performance range 2305–2385 nm located four rows from the cloud spectra.

## 9   Conclusions

We have developed and applied a method to characterize and correct stray light in a push-broom spectral imager. The stray-light distribution in measurements with the TROPOMI-SWIR instrument has been characterized as a function of the position of the main light beam on the detector. A detailed description was possible using a monochromatic quasi-point light source.

By combining saturated and unsaturated measurements, the dynamic range of the signal was extended to more than seven orders. A fast correction algorithm has been devised and implemented in the operational data processor, based on Van Cittert deconvolution. The stray light is approximated by a far-field kernel independent of position and wavelength and an additional kernel representing the main reflection. A fraction of 4.4 % of the detected light is stray light that is transferred back to the correct detector positions. As a result, a reduction of the stray-light signal in for example a simulated dark forest scene close to

bright clouds by a factor of about 10 has been demonstrated. Simulations indicate that this brings the stray-light error in gas-column retrievals within the required budget. In addition, the instrument contains five SWIR diode lasers that enable long-term, in-flight monitoring of the stray-light distribution.

### Data availability

The underlying data of the figures presented in this publication can be found at ftp://ftp.sron.nl/open-access-data/pault.

**Appendix A: Position of the main reflection**

The reflection kernel and the correction algorithm have been designed together to position the main reflection correctly.

If the peak is centred at column $c_0 = c_{\alpha,\lambda}$, there is a reflection at column $c_{\text{refl}}$. Distance $c_{\text{refl}} - c_0$ is constant (and almost zero). If the image is shifted by $c_{\text{centre}} - c_0$ with arbitrary fixed column index $c_{\text{centre}}$, the peak is found at $c_{\text{centre}}$ and the reflection a fixed column distance away. The choice of $c_{\text{centre}}$ is arbitrary.

If the peak is centred at row $r_0 = r_{\alpha,\lambda}$, the reflection is at row $r_{\text{refl}}$. They are at the same distance from fixed row index $r_{\text{mirror}}$, but on opposite sides:

$$r_0 - r_{\text{mirror}} = r_{\text{mirror}} - r_{\text{refl}}. \tag{A1}$$

This means

$$r_{\text{refl}} = 2r_{\text{mirror}} - r_0. \tag{A2}$$

If the image is shifted by $r_0 - r_{\text{centre}}$ with arbitrary fixed row index $r_{\text{centre}}$, the peak is found at $2r_0 - r_{\text{centre}}$ and the reflection at

$$r'_{\text{refl}} = 2r_{\text{mirror}} - r_{\text{centre}}, \tag{A3}$$

where $r'_{\text{refl}}$ is independent of original peak position $r_0$. An obvious choice for $r_{\text{centre}}$ would be $r_{\text{centre}} = r_{\text{mirror}}$, but this row index would have to be determined first by fitting. Parameter $r_{\text{centre}}$ is set to 127.5, exactly at the middle of the detector.

Skipping some details on signal scaling and removal of the main peak, the reflections in all frames after shifting are averaged to form the reflection kernel. This kernel may be cropped, but the middle should be at $(r_{\text{centre}}, c_{\text{centre}})$; the reflection is not at the middle of the kernel, but at a row distance

$$r'_{\text{refl}} - r_{\text{centre}} = 2(r_{\text{mirror}} - r_{\text{centre}}) \tag{A4}$$

from the middle.

In the correction algorithm, any input frame is mirrored relative to row index $r_{\text{centre}}$. With our choice, this is simply reversing the row order, when the frame has 256 rows. The row index of the peak becomes

$$r''_0 = r_{\text{centre}} - (r_0 - r_{\text{centre}}) = 2r_{\text{centre}} - r_0. \tag{A5}$$

Afterwards, the frame is convolved with the reflection kernel. This means the peak at row index $r''_0$ will produce a secondary peak at a distance given by Eq. (A4), at row index

$$r''_{\text{refl}} = r''_0 + 2(r_{\text{mirror}} - r_{\text{centre}}). \tag{A6}$$

Using Eqs. (A5) and (A2), this is the same as

$$r''_{\text{refl}} = 2r_{\text{mirror}} - r_0, \tag{A7}$$

$$= r_{\text{refl}}, \tag{A8}$$

i.e. the secondary peak is created at the correct position of the reflection. The result is subtracted from the input frame affected by the reflection (without mirroring). There are several advantages of this method: $r_{\text{mirror}}$ and $r_{\text{refl}}$ do not have to be known, the implied $r_{\text{mirror}}$ does not have to be an integer and the correction algorithm is relatively fast.

*Author contributions.* The algorithm and experiments were designed by RS, MK, PT and Airbus Defence and Space Netherlands. The experiments were implemented and executed by Airbus Defence and Space Netherlands and KNMI with support of SP, MK, PT, SC and RvH. The calibration data were derived by SC and PT. The data analysis and simulations were performed by PT. RH and IA supervised the study. All authors discussed the results and commented on the manuscript.

5  *Competing interests.* The authors declare that they have no conflict of interest.

*Acknowledgements.* The authors would like to thank the teams of Airbus Defence and Space Netherlands and KNMI for organizing the calibration campaign and in particular the operators for the tireless data acquisition. TROPOMI is a collaboration between Airbus Defence and Space Netherlands, KNMI, SRON and TNO, on behalf of NSO and ESA. Airbus Defence and Space Netherlands is the main contractor for the design, building and testing of the instrument. KNMI and SRON are the principal investigator institutes for the instrument. TROPOMI

10  is funded by the following ministries of the Dutch government: the Ministry of Economic Affairs, the Ministry of Education, Culture and Science, and the Ministry of Infrastructure and the Environment.

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
