# Peer review of "Characterization and correction of stray light in TROPOMI-SWIR"

_Atmospheric Measurement Techniques, 2017_

## Referee Comment (RC1) · Anonymous Referee #1 · 12 Feb 2018

Characterization and correction of stray light in TROPOMI -SWIR
Paul J. J. Tol et al.
AMT-2017-455

Summary Comment

This is an important paper bench-marking the performance of the TROP-OMI instrument in the SWIR band. The results of the laboratory testing are significant in their use in reducing on-orbit observations and test data. The paper should be published. However, the methodology is complex and the information provided is not sufficient for an outside evaluation to adequately assess the success of the test analyses reported. The reported performance of the instrument is not clearly stated, as different presentations of the information lead the reader to form different conclusions: the raw corrected data, the plotted slit function behaviour and the corrected data. These seem to range from a dynamic range of 1.0E-7 to 1.0E-4 or 1.0E-5. The different presentations and their implications should be clearly described to the reader.

General comments

In some 2-D detector systems it is very difficult to separate optical stray light effects from artifacts generated by the readout and resetting operations. There should be some discussion added to address these issues in order that the stray light effects documented are considered in the correct context. 1.0E-7 as a far-wing stray light rejection is good performance for a double monochromator. Accepting that kind of performance in a single monochromator with a array detector should be done with the greatest care.

The description of the testing and analysis, and especially the attribution of systematic affects in section 3, is not likely to be accessible to a wider audience. The terms and methods for drawing the conclusions presented should be more fully explained. In particular, separating blooming and resetting from the effects of stray light is an important issue.

It might be instructive to illuminate more than one pixel in a 'two-source' test to examine the separation of stray light effects and effects from other sources such as blooming and charge carry-over (incomplete pixel resetting).

Some detailed comments listed by PDF page number.

Page 1 Line 1          Suggest: "... Short-Wave InfraRed ..."

Page 1 Line 3          Suggest: "... are needed that are minimally contaminated by instrumental stray light."

Page 1 Line 5          "... seven orders of magnitude by making measurements with some saturated and some unsaturated detector pixels ..."

Page 1 Line 7          "... signal, for example in a dark forest scene close to bright clouds, by ..."

| | |
|---|---|
| Page 1 Line 8 | "It is expected that this reduces the stray-light error sufficiently for accurate gas-column retrievals." Can a stronger statement be made? |
| Page 1 Line 10 | ".. long-term, in-flight ..." |
| Page 1 Line 12 | Suggest: "... (TROPOMI) is the only instrument on board the ..." |
| Page 1 Line 16 | Suggest: "... sampling interval of 0.1 nm." |
| Page 1 Line 20 | "To achieve the required accuracy of the measurement of spectral radiance, an accurate correction for the stray light must be included in the analysis." |
| Page 2 Line 4 | "Examples of corrected measurements are..." |
| Page 2 Line 7 | If possible, a simplified diagram of the optics would be welcome to readers. Describe what is meant by an 'immersed grating'. |
| Page 2 Line 13 | The methodology for reading out the detectors should be described. Is it CCD, random addressed, multiple amplifiers, A/D(s) etc.? |
| Page 2 Line 17 | "... calibration were performed ..." |
| Page 2 Line 32 | "... shutter in front of the spinning mirror ..." |
| Page 3 Line 5 | "... the instrument was reduced ...". This description of the attenuation and gain setting indicates that all but the shortest integration time will be saturated. It is suggested that the reason for this be stated explicitly (i.e.: the sensitivity needed to measure the weak stray light on remote pixels) and some indication of the results of tests done to show that the signals are from stray light and not electronic artifacts. |
| Page 3 Line 12 | The methodology here seems strange. One would expect that the dark signal would be a minimum at the smallest integration time and grow larger at the longest integration time. The analysis described here seems to be neglecting dark count and concerning itself only with 'background' whatever that encompasses. Perhaps some more detail here would be appropriate. There are a number of phenomena which must be characterized or shown not to be important: electronic readout noise, charge carry-over from reset (and any spreading to other pixels) and actual dark count (thermal generation). |
| Page 3 Line 23 | 'absolute signal' should be defined |
| Page 4 | Figure 2. "Background-corrected light peak at different exposure times: (a) |

0.2ms, (b) 4.6ms, (c) 106ms, (d) 1998ms. Only at the shortest exposure time are there no pixels saturated."

| | |
|---|---|
| Page 4 Line 8 | The quantity proportional to intensity after the integration time is taken into account is probably proportional to current but more likely presented in terms of A/D counts per second. The wording should be made more precise to make the information clear to the reader. |
| Page 4 Line 9 | "...the dynamic range of the signal current is *theoretically* larger than seven orders of magnitude." This does not automatically mean stray light can be characterized to that level. The later figures suggest that the stray light is on the order of 4 orders below the peak intensity at distant pixels. The discrepancy between these two numbers - if the background subtracted includes effects additional to stray light - may lead to a non-linearity in the stray light correction process. |
| Page 4 Line 12 | "... saturated by light, not dark current .." It is not clear what this is to mean given that the generation of electrons by light and by heat is essentially the same. |
| Page 4 Line 13 | It is not clear what correction has generated the change from figure 2 to figure 3. It suggests a very effective reduction of stray light signal. However, this must be taken in the context of the earlier comments about line 9. It would be helpful if the correction procedure were described fully and more clearly. |
| Page 5 Line 4 | The unknown agent leading to alternating values of signal level is somewhat concerning. One question here is whether a study has been done to see if all combinations of pixel readout values are statistically likely. A problem with the analog and A/D circuitry can cause strange effects in the observed measurements. Perhaps a comment about this might be a useful addition. In the RETICON there is a potential for this kind of effect because of the way charge is shared between pixels. The gain pattern can be seen where it is clear that small differences in the mask used to define the electrodes causes the issue. |
| Page 5 Line 5 | "Second, in the conversion to signal current, exposure time 0.2 ms is actually replaced by 0.14 ms." Is this a statement that measurements were done again at a different integration time or is this the determination of a corrected integration time for measurement nominally made at 0.2 ms? Does this perhaps feed into the high bias of the model stray light cureve relative to the data? |
| Page 5 Line 11 | It is not clear to this reviewer what is mean by "... Gaussian and block distribution." |

| | |
|---|---|
| Page 8 Line 11 | "(across-track and along-track)".  This description seems somewhat misleading.  The detector has cross-track imaging but the other direction is spatial.  Better to leave the "along-track" description out to avoid confusion. |
| Page 10 Line 15+ | The equations were very difficult to read because of problems with formatting in the PDF.  A description of the methodology which is more accessible to a wider audience would be a great addition to the paper.  It is left to the authors to ensure that the equations are correct. |
| Page 7 Line 9 | Is the 2.1 spread compared to the 1.1 degree source in agreement with the instrument design model?  It would be more informative to convert the 1.1 degrees to the expected pixel size.  It is just quibbling,nn but the bump reported in the next line looks more like 0.6 or 0.7 %. |
| Page 7 Lin 10 | The comment concerning the Xe lamp should be explained.  It is not clear why once source should spread spatially while the other does not. |
| Page 8 Line 2 | "... at the peak positions used ..."  "The ISRF was determined ..." |
| Page 8 Line 6 | "... contain more stray light than any diagonal cross section .."  Can electronic contributions be ruled out? |
| Page 8 Line 8 | "... Once the general behaviour of the stray light was examined, ..." |
| Page 8 Line 11 | "the point response function (PRF)" "point spread function (PSF)" is more commonly used. |
| Page 19 Line 5 | "We have developed and applied a novel method ..."  While the work presented is thorough and makes use of a very useful algorithm, this may be overstating the situation somewhat... |

---

## Referee Comment (RC2) · Anonymous Referee #2 · 30 Mar 2018

General Comments:

This paper presents the laboratory characterization of stray light for the TROPOMI SWIR spectrometer, and the development of a correction algorithm to remove the stray light. The stray light is characterized using a monochromatic quasi-point light source; the signal is obtained with more than 7 orders of magnitude by combining saturated and unsaturated measurements taken at different exposure times. Analysis of the spectral-spatial spread function (SSSF), indicates that the stray light can be approximated by the far field of a stable kernel and a main reflection kernel, both of which do not vary with wavelength and across-track position. A correction algorithm is developed to correct stray light in in-flight observations. Application of the stray light correction to stray light measurements as well as synthetic earthshine observations show that the stray

light can be significantly reduced. This study presents key information about TROPOMI SWIR and its performance and is very suitable for publication in AMT. This paper is generally well organized, and the methodology and results are generally well described. However, one of the important conclusions "this reduces stray light error sufficiently for accurate gas-column retrievals" is not supported by details. It does not say if the developed correction algorithm is the operational algorithm or will be implemented for operational processing. If it is not the operational algorithm (i.e., described in ATBD). The authors should talk about it in the introduction and if possible compare the performance between this correction algorithm and the operational algorithm. Some of the sentences are difficult to understand and the English writing needs to be improved. The abstract and conclusion can be improved to include more important aspects of this study. Overall, I think that this paper can be published after addressing the comments mentioned here and specific comments below.

Specific Comments:

1. In abstract, L3, the sentence needs to be rephrased to make it read well. You can change it to "For this purpose, it is required to have calibrated radiance measurements in which . . ."

2. In abstract, L6, suggest adding "from different exposure times" after "unsaturated measurements"

3. In abstract, L6-8, the sentences do not read well. You may add a little more detail about the algorithm, including the approximation representation with a far-field stable and reflection kernels. Suggest changing the order of sentences to something like " . . . unsaturated measurements. Analysis of the stray light indicates about 4.4% of the detected light is correctable stray light. An algorithm was then devised . . . . . . Applying this correction significantly reduces the stray-light signal for example . . ."

4. P1, L15, change "distance" to "interval" as it is more frequently used for the spectral dimension

5. P2, L7, if it is a common telescope, should it be called as "SWIR telescope"? Suggest changing it to "telescope"

6. P3, first paragraph, it is not clear about why to avoid saturation at the shortest exposure time, and why fewer frames are averaged at longest exposure time. Based on section 3, the shortest exposure is to measure the strong signal while the longest exposure time is to measure the weakest signal in the wing of point spread function. L3-5 of page 4 can be moved here to help understand the purpose of taking measurement at 4 exposure times.

7. P3, L13-14, change to "ensures that no important features only closer to specific wavelengths or swath angles are missed"

8. P3, L15, change to "background measurements"

9. P3, L17, add "as background measurements" after "is used"

10. P3, L18, suggest changing to "The peak in a given detector area only occurs in a small subset of the light measurements, . . ."

11. P3, L20, suggest "at which" to "and", because "the longest exposure time of 1998 ms", meant by "which" does not occur before "at which"

12. P4 L6, suggest changing to ". . . does not saturate a pixel (a saturated signal . . .)"

13. P4, L8-9, suggest changing to "the signal can be measured with a dynamic range of more than seven orders of magnitude"

14. P4, L11-12, suggest changing to "the signal of unsaturated pixels becomes too high due to spilling from a direct neighbor pixel saturated by light, not dark current"

15. P5, L10-11, agree with the first reviewer, the sentence of "The peak positions and . . .. Gaussian and a block distribution in these two dimensions . . ." is not clear

16. P5, L13, change to "After discarding"

17. P5, last paragraph in Sect. 3, do the last four sentences describe the PRNU correction as it is not clear to me. If not, it is useful to describe the PRNU in a few sentences.

18. P9, L2-3, it is not clear why only stray light originating at spatial coordinates is corrected. Does it mean that the spectral stray light is not corrected? Please clarify it.

19. P10, L4, change "using the element-wise product o" to "where o is the element-wise product"

20. P10, L21, add "is as follows" after "The rationale of this algorithm" as it is not a sentence and change "which is" to "and is"

21. P11, L1, change to ' . . . center. The iteration consists of . . ."

22. P11, L7, change to ", as explained"

23. P15, L8, suggest adding "After the stray light correction" before "Some stray light . . ." to make it more readable.

24. P16, Figure 14 caption, change "measurement" to "measurements"

25. P16, the last paragraph that continues on P18 is not easy to understand and seems to distract the flow from Fig. 15a to Fig. 15b. I think that the main purpose is to describe how to construct signal with stray light by convolving Fig 15 a with the SSSF. It is not clear to me about the need to talk about merged frame and conversion to kernels here. Please make the description clearer.

26. Figure 16 caption, suggest changing to ". . . respectively, for the forest spectrum in 2305-2385 nm located four rows from the cloud spectra"

27. P18, L9-10, it is not clear about "one merged frame per 20 pixels". Do you mean merging 20 pixels across the track (i.e., merging 20 rows)?

28. P19, can you provide more quantitative details to support the last sentence of

section 8 as this is an important conclusion in abstract and conclusion and the residual error is still up to 1%. In addition, has the instrument noise been added to the simulation? Will the inclusion of instrument noise affect the performance of stray light correction?

29. Also, it is not clear if this stray light correction algorithm is the operational algorithm to be used in level 0 to 1b processor (i.e., described in ATBD)? It looks like it is not, then how is the performance and speed of this correction compared to those of the operational correction?

30. In conclusion, it is useful to expand a little bit about the main idea of the fast correction algorithm (e.g., stable kernel, reflection kernel etc.)

31. P20, L2, this sentence does not read well, suggest changing to "The fitted r_center is 127.5, exactly . . ."

———————————————————

---

## Referee Comment (RC3) · Anonymous Referee #3 · 23 Apr 2018

*Characterization and correction of stray light in TROPOMI-SWIR, Paul J.J. Tol et al., MS No.: amt-2017-455,*

**General comments**

Initial paragraph or section evaluating the overall quality of the discussion paper.

The paper is well written and of good quality, with a considerable number of new interesting topics and techniques related to stray light characterization and correction, and shall certainly be published. However, I am of the opinion that the quality of the paper can be much improved to be more useful with a comparatively small additional effort, in line with the comments and suggestions provided below. After these comments and suggestions have been adequately addressed, the paper shall certainly be published.

1. For the applications in TROPOMI SWIR it is essential to express, quantify and present stray light at L0 and L1b as a percentage of the useful signal for realistic earth atmosphere low-albedo scenes and signals within absorption peaks. This is currently not the case. For example in figure 14 it can be seen that the difference between bright and dark scenes is roughly a factor 8 (see also page 15, line 14), and the difference between rather deep absorption line and continuum is also roughly about a factor 8. Both together would make a difference of a factor 64. For example, in the legend of figure 15 (but also in other places in the paper) it is clear that the stray light is expressed as percentage of the expected continuum in the given row, not as percentage of the useful signal in the absorption lines. If the authors prefer to present the stray light like this, it is essential to at least also show the stray light as percentage with respect to the low albedo useful signal in the absorption lines. This shall be added to the paper in the text, figures and conclusions.

**Specific comments**

Section addressing individual scientific questions/issues.

1.
Page 8, figure 7:
It is recommended to (also) show the row distances similarly as the column distances as shown in figure 8, to allow a better comparison between the two. For example, it is recommended to add a figure like the lower figure in figure 7 at a horizontal scale of +/-100 pixels, to allow better comparison with figure 8.

2.
Page 11, line 20:
The number is given as 4.3% of the detected light.
In line with the general comment given above, it is recommended to provide also the percentage numbers with respect to low-albedo numbers in the absorption lines.

3.
Page 18, lines 7-10:
Again, in line with the above, this is why it is important to calculate the stray light fraction with respect to the useful signal, because the absorption lines and low-albedo scenes will be filled in with higher signals from the continuum and the higher-albedo scenes. This may also affect the instrument spectral response functions. Please reflect this in the text.

4.
Page 18, lines 11-12:
Again, in line with the above, this is as written not agreed. This sentence shall be removed. In case the authors want to keep this sentence, another sentence needs to be added explaining why stray light needs to be assessed as fraction with respect to the useful signal, not the signal in the continuum.

5.
Section 9, conclusions.

The statement "It is expected that this brings the stray-light error in gas-column retrievals within the required budget" is not discussed or supported by analyses in the paper. Either provide more support material in the paper to corroborate this statement, or remove it from the conclusions.

**Technical corrections**

Compact listing of purely technical corrections (typing errors, etc.).

1.
The legends of (almost) all figures have symbols that cannot be read. This is at least true in the pdf version. Please correct.

2.
(Almost) all equations (e.g. 3,4,5,7,8,9,10) have symbols that cannot be read. This is at least true in the pdf version. Please correct.

3.
Figure 15: In the legends, please indicate if this is a fraction, percentage or something else.

---

## Author Comment (AC1) · 18 May 2018

**Author comment on "Characterization and correction of stray light in TROPOMI-SWIR" by Paul J. J. Tol et al., manuscript amt-2017-455, Anonymous Referee #1**

We would like to thank Referee #1 for the comments to improve our manuscript. In this document, we provide our reply to the comments. The original comments made by the referee are numbered and typeset in red. Page, line and figure numbers refer to the old version of the manuscript. After the reply we provide a revised version of the manuscript, in which all changes are highlighted.

1. *Summary Comment.* This is an important paper bench-marking the performance of the TROP-OMI instrument in the SWIR band. The results of the laboratory testing are significant in their use in reducing on-orbit observations and test data. The paper should be published. However, the methodology is complex and the information provided is not sufficient for an outside evaluation to adequately assess the success of the test analyses reported. The reported performance of the instrument is not clearly stated, as different presentations of the information lead the reader to form different conclusions: the raw corrected data, the plotted slit function behaviour and the corrected data. These seem to range from a dynamic range of 1.0E-7 to 1.0E-4 or 1.0E-5. The different presentations and their implications should be clearly described to the reader.

**Adjusted.** Extra information is given on the methodology and tests (see items 16, 19, 20, 26, 31 and 35 below). The confusion around the performance using stray-light characterization data, corrected example data and slit function data is discussed in items 24 and 28.

2. *General comments.* In some 2-D detector systems it is very difficult to separate optical stray light effects from artifacts generated by the readout and resetting operations. There should be some discussion added to address these issues in order that the stray light effects documented are considered in the correct context. 1.0E-7 as a far-wing stray light rejection is good performance for a double monochromator. Accepting that kind of performance in a single monochromator with a array detector should be done with the greatest care.

**Adjusted.** The readout is discussed in items 16, 20 and 27. The orders of magnitude are discussed in item 24. Two line features in the stray light are discussed in item 35.

**3.** The description of the testing and analysis, and especially the attribution of systematic affects in section 3, is not likely to be accessible to a wider audience. The terms and methods for drawing the conclusions presented should be more fully explained. In particular, separating blooming and resetting from the effects of stray light is an important issue.

Adjusted. The description is extended in items 20, 23, 25 and 29.

4. It might be instructive to illuminate more than one pixel in a 'two-source' test to examine the separation of stray light effects and effects from other sources such as blooming and charge carry-over (incomplete pixel resetting).

Adjusted. Stray light can only be measured with the full spectrometer and that is no longer available for testing. However, measurements have been taken where many pixels have been illuminated and at regular intensities. The stray-light correction still works. See item 24.

Some detailed comments listed by PDF page number.

5. Page 1 Line 1 Suggest: "... Short-Wave InfraRed ..."

**Not adjusted.** According to the AMT house standards, a capitalized abbreviation does not necessarily warrant the capitalization of the written-out form and non-standard usage of capitalization is only acceptable for proper nouns.

6. Page 1 Line 3 Suggest: "... are needed that are minimally contaminated by instrumental stray light."

Adjusted according to the suggestion.

7. Page 1 Line 5 "... seven orders of magnitude by making measurements with some saturated and some unsaturated detector pixels ..."

**Adjusted.** The suggestion does not make it clear that the same setup is measured in different ways. The text is changed to: "... seven orders of magnitude by performing measurements with different exposure times, saturating detector pixels at the longer exposure times."

8. Page 1 Line 7 "... signal, for example in a dark forest scene close to bright clouds, by ..."

**Adjusted.** The first comma is added to the text, but the second comma would change the meaning; the factor 10 reduction is not fixed, but applies to the difficult forest scene. The text is changed to: "Applying this correction significantly reduces the stray-light signal, for example in a simulated dark forest scene close to bright clouds by a factor of about 10."

9. Page 1 Line 8 "It is expected that this reduces the stray-light error sufficiently for accurate gas-column retrievals." Can a stronger statement be made?

Adjusted. A full retrieval analysis with statistics over different scenes using the final retrieval algorithm is beyond the scope of this article. The given conclusion is based on one challenging scene type and a non-scattering retrieval algorithm. At the end of Sect. 8 extra information is given, but the numbers are not general enough (e.g. only one scene) to include them also in the abstract or conclusion. The old text in Sect. 8 is: "Calculations have shown that the error in the retrieved gas columns is within the allowed error budget." This is replaced by: "Applying non-scattering retrieval (without aerosol or cirrus parameters) to the forest spectrum used for Fig. 16, the error of the methane column reduces from 14 % before correction to 0.26 % after correction, within the error budget of 0.35 % for stray light. The error of the given CO column of 3.0 %. The forest spectra at other rows give similar results after correction." The abstract and conclusion change from old "It is expected that ..." to new "Simulations indicate that ...".

10. Page 1 Line 10 ".. long-term, in-flight ..."

Adjusted according to the suggestion.

11. Page 1 Line 12 Suggest: "... (TROPOMI) is the only instrument on board the ..."

Adjusted according to the suggestion.

12. Page 1 Line 16 Suggest: "... sampling interval of 0.1 nm."

Adjusted according to the suggestion.

13. Page 1 Line 20 "To achieve the required accuracy of the measurement of spectral radiance, an accurate correction for the stray light must be included in the analysis."

**Adjusted.** The term 'analysis' is replaced by 'data processing' because the correction is applied from level 0 to 1, not 1 to 2. The text is changed to: "To achieve the required accuracy of the spectral-radiance measurements, an accurate correction for the stray light must be included in the data processing."

14. Page 2 Line 4 "Examples of corrected measurements are..."

Adjusted according to the suggestion.

15. Page 2 Line 7 If possible, a simplified diagram of the optics would be welcome to readers. Describe what is meant by an 'immersed grating'.

**Adjusted.** A diagram is included in the article (see Fig. 1 in this author comment). The immersed grating is described by adding the text: "The immersed grating consists of a silicon prism as the immersive medium with a diffraction grating on one surface. By illuminating the grating from inside the prism, the resolving power is increased by the refractive index of silicon. This allows the spectrometer to be much smaller than with a conventional echelle grating."

**Figure 1.** The setup for the stray-light measurements. The elements after the OPO are: neutral density filter F, shutter S, fold mirrors FM1 and FM2, spinning mirror SM, integrating sphere IS, field stop P, parabolic mirror PM and window W of the vacuum chamber containing the TROPOMI instrument.

16. Page 2 Line 13 The methodology for reading out the detectors should be described. Is it CCD, random addressed, multiple amplifiers, A/D(s) etc.?

Adjusted. The text already mentions the detector is "based on silicon complementary metal-oxide-semiconductor (CMOS) technology", i.e. it is not CCD. At the first mention of the detector (page 2, line 10), the model name is added to the text: "(Saturn model by Sofradir, France)". Extra information on the signal conversion and storage is added to the text (page 2, line 12): "In each detector pixel, the signal charge is converted into a voltage by a capacitive transimpedance amplifier (CTIA) and after a given exposure time stored in a sample-and-hold circuit before read-out. Details of the detector read-out and characterization are given by Hoogeveen et al. (2013)." An implication of the CTIAs is given when discussing blooming (page 4, line 11): "Due to the use of a CTIA in each detector pixel, the signal does not affect the detector bias voltage and signals of neighbouring pixels do not affect each other, unless a signal saturates. In that case blooming occurs, seen as a ring around the peak...".

Part of the referenced detector characterization was the optimization of the instrument timing. The integration capacitor is discharged (resetted) long enough to minimize the corresponding memory effect and non-linearity. The charge on the integration capacitor is sampled long enough to minimize the corresponding memory effect. The remaining memory is taken care of by discarding the first two frames at a given exposure time and illumination pattern. The last frame is also discarded in case its readout is affected by the following measurement, which takes place at the same time.

17. Page 2 Line 17 "... calibration were performed ..."

Adjusted according to the suggestion.

18. Page 2 Line 32 "... shutter in front of the spinning mirror ..."

Adjusted according to the suggestion.

19. Page 3 Line 5 "... the instrument was reduced ...". This description of the attenuation and gain setting indicates that all but the shortest integration time will be saturated. It is suggested that the reason for this be stated explicitly (i.e.: the sensitivity needed to measure the weak stray light on remote pixels) and some indication of the results of tests done to show that the signals are from stray light and not electronic artifacts.

**Adjusted.** The textual suggestion is incorporated as "... the instrument is reduced ...". The reason for the saturation is stated explicitly by moving the explanation at the start of the next section (page 4, line 2-5) to this section. The term 'gain setting' in the comment prompts another clarification: only the exposure time is changed between measurements, not a gain setting or other detector setting. One indication that the signals are from stray light and not electronic artifacts is given in Sect. 8, where the stray-light correction is applied accurately to measurements with much lower intensities. The text has been changed to: "... taken at four different exposure times with about 20-fold increment steps (0.2, 4.6, 106 and 1998 ms), without changing any other setting. With a neutral density filter just after the OPO, the spectral radiance at the instrument is reduced to 200 times the highest value in nominal operations. At this radiance, the detector is never saturated at the shortest exposure time. However, the signal with this exposure time is noisy away from the peak. At the three longer exposure times, the signal has a much better signal-to-noise ratio away from the peak, but the peak is

saturated. These three exposure times are used to increase the dynamic range of the image by four orders of magnitude (see Sect. 3)."

**20.** Page 3 Line 12 The methodology here seems strange. One would expect that the dark signal would be a minimum at the smallest integration time and grow larger at the longest integration time. The analysis described here seems to be neglecting dark count and concerning itself only with 'background' whatever that encompasses. Perhaps some more detail here would be appropriate. There are a number of phenomena which must be characterized or shown not to be important: electronic readout noise, charge carry-over from reset (and any spreading to other pixels) and actual dark count (thermal generation).

Adjusted. The text is adapted to explain the background better. The first time 'background' is mentioned (page 2, line 32), the text is already "Background measurements are taken by closing a shutter in front of the spinning mirror." As a clarification, the following is added after this text: "Background data contain the same pixel-dependent offset, detector dark current and thermal background as light data taken at the same exposure time." When the background is used (page 4, line 7), the text is expanded to: "The background signal of this pixel at the chosen exposure time is subtracted, removing the pixel-dependent offset, detector dark current and thermal background."

The dark signal is indeed larger at longer exposure times, but it is included both in the light measurements and in the background measurements taken at the same exposure time, so it is accurately removed by the background subtraction. Due to the use of CTIAs, the charge spreading to other pixels is not an issue when the signal is not saturated. This is added to the text in item 16. When the signal is saturated, blooming occurs, which is described later on with examples (page 4, line 11). Text is added regarding the electronic readout noise when whole merged frames are shown in Sect. 4 (page 5, line 15): "The electronic noise has been suppressed to about  $10^{-7}$  times the peak signal. It determines the lower end of the dynamic range." See also item 24.

21. Page 3 Line 23 'absolute signal' should be defined

Adjusted. The text is changed to: "... the total signal (due to external light, thermal background and detector dark current) ..."

22. Page 4 Figure 2. "Background-corrected light peak at different exposure times: (a) 0.2ms, (b) 4.6ms, (c) 106ms, (d) 1998ms. Only at the shortest exposure time are there no pixels saturated."

Adjusted. The new text is: "Only at the shortest exposure time no pixels are saturated."

23. Page 4 Line 8 The quantity proportional to intensity after the integration time is taken into account is probably proportional to current but more likely presented in terms of A/D counts per second. The wording should be made more precise to make the information clear to the reader.

**Adjusted.** The data is actually in electrons per second, but the unit is not important. The text is changed to make a distinction between signal rate (anything per unit time) and signal current (something to do with electrons): "The result is divided by the exposure time to get a signal rate in digital counts per second. By applying a separate calibration of the number of electrons per count, it is converted to a signal current."

24. Page 4 Line 9 "...the dynamic range of the signal current is *theoretically* larger than seven orders of magnitude." This does not automatically mean stray light can be characterized to that level. The later figures suggest that the stray light is on the order of 4 orders below the peak intensity at distant pixels. The discrepancy between these two numbers -if the background subtracted includes effects additional to stray light -may lead to a nonlinearity in the stray light correction process.

**Adjusted.** The dynamic range of the signal current in the merged frames is also in practice seven order of magnitude. This is seen in Fig. 4, where the noise is at  $10^{-7}$  and the peak at 1. In the median of all frames for the stable kernel, the noise is actually smaller. The *characterization* is performed with seven orders, but the *correction* is worse (although good enough), due to neglecting small features. These are discussed in Sect. 4. The noise in the calibration measurements for stray light is negligible in comparison to noise in normal measurements. The later figures referred to in the comment are

presumably the examples of normal measurements in Fig. 12b and c. Here the noise in the far field of the measurements is at  $10^{-5}$  and  $10^{-4}$ , respectively, because the light intensity was much smaller, no merged frames were used and hence the signal-to-noise ratio was much smaller. This is not due to stray light or its correction. To clarify this point, text is added near Fig. 12 (page 15, line 8): "The measurements for Fig. 12b and c were performed with lower light intensities, leading to a smaller signal-to-noise ratio than available in the stray-light characterization. In these cases with illumination over many pixels and at regular intensities, the stray-light correction works well."

25. Page 4 Line 12 "... saturated by light, not dark current .." It is not clear what this is to mean given that the generation of electrons by light and by heat is essentially the same.

**Adjusted.** The generation of electrons by light and by heat should indeed be the same, but the data show otherwise. Maybe the dark current of these bad pixels only seems large due to a defective circuit. This was not investigated, because it is rare and does not affect the production of the stable kernel. Text has been added (page 4, line 13): "The reason why blooming does not occur due to dark current is unknown and not investigated further as there are only about 80 detector pixels with a large enough dark current."

26. Page 4 Line 13 It is not clear what correction has generated the change from figure 2 to figure 3. It suggests a very effective reduction of stray light signal. However, this must be taken in the context of the earlier comments about line 9 (item 24). It would be helpful if the correction procedure were described fully and more clearly.

**Adjusted.** The transformation from Fig. 2 to Fig. 3 is fully described by the first paragraph of Sect. 3. It is just selection of signals, subtraction of the corresponding background and division by the corresponding exposure time; there is no (other) correction. This may already have been clarified by the extra text on background (item 20), signal rate (item 23) and dynamic range (item 24). Figures 2 and 3 show different data types, so the values should not be compared directly. In case this is not clear, text is added to the caption of Fig. 3: "Combination of data in Fig. 2 (from signal to signal current) to increase the dynamic range...".

27. Page 5 Line 4 The unknown agent leading to alternating values of signal level is somewhat concerning. One question here is whether a study has been done to see if all combinations of pixel readout values are statistically likely. A problem with the analog and A/D circuitry can cause strange effects in the observed measurements. Perhaps a comment about this might be a useful addition. In the RETICON there is a potential for this kind of effect because of the way charge is shared between pixels. The gain pattern can be seen where it is clear that small differences in the mask used to define the electrodes causes the issue.

**Not adjusted.** The text already states that it "is not understood" (page 5, line 3) and speculation seems not helpful. As far as we know, the RETICON shares charge that is generated between pixels differently depending on the signal, because the bias voltage depends on the signal. This is not the case for this detector, due to the fact that the CTIA maintains a stable bias voltage (see item 16). Fixed gain variations between pixels are not relevant, because the alternate readings are taken at the same pixel. Statistics are not needed because the variation between signals is not subtle: it can be larger than 100 digital counts, but only at an exposure time of 0.2 ms and when the signal is very high. Non-linearity of the A/D circuitry has been investigated, but the effect is negligible.

28. Page 5 Line 5 "Second, in the conversion to signal current, exposure time 0.2 ms is actually replaced by 0.14 ms." Is this a statement that measurements were done again at a different integration time or is this the determination of a corrected integration time for measurement nominally made at 0.2 ms? Does this perhaps feed into the high bias of the model stray light curve relative to the data?

**Adjusted.** The measurements were not performed again but the value of the exposure time was changed in the calculation. The text was changed to: "Second, in the conversion to signal current, nominal exposure time 0.2 ms is actually replaced by the value 0.14 ms."

The curve in Fig. 8b is not a stray-light model but an ISRF model, based on separate calibration data that did not use saturation. It is accurate on a sub-pixel scale, but only in the central 9 pixels around the peak and accurate to about 0.001. Here it is extrapolated very far, just for comparison. The ISRF model is not used in the stray-light correction. The

extrapolation is already mentioned in the caption, but the curve label is changed from 'ISRF' to 'ISRF, extrapolated' and the main text is changed from (page 8, line 3) "The logarithmic plot shows the ISRF over an extended range, but in trace-gas retrievals it is only used in the range over the central 9 pixels." to "It is defined over the central 9 pixels that is used in trace-gas retrievals, but in the logarithmic plot the ISRF is extrapolated beyond that range."

**29. Page 5 Line 11 It is not clear to this reviewer what is mean by "... Gaussian and block distribution."**

Adjusted. Block distribution is meant as a more descriptive, alternative name for uniform distribution, but it is clearer to include the fit function explicitly. The original text is: "The peak position and the integrated signal current are determined in each frame by fitting the convolution of a Gaussian and a block distribution in two dimensions. The variation of the light intensity is then removed by dividing each frame by the corresponding integrated signal current." The new text is: "At this stage, a merged frame consists of signal current  $I_{\text{meas}}[r,c]$  as a function of detector row r and column c. The area near the peak is fitted with a two-dimensional function S(r,c). The one-dimensional convolution of a Gaussian distribution with mean 0 and full width w is given by

$$\mathcal{B}(x;\sigma,w) = \frac{1}{2w} \left[ \operatorname{erf}\left(\frac{x+w/2}{\sqrt{2}\sigma}\right) - \operatorname{erf}\left(\frac{x-w/2}{\sqrt{2}\sigma}\right) \right].$$
(1)

The fit function is

$$S(r,c) = a \mathcal{B}(r - r_0; \sigma_{\text{spat}}, w_{\text{spat}}) \mathcal{B}(c - c_0; \sigma_{\text{spec}}, w_{\text{spec}}),$$
(2)

with spatial (vertical) peak position  $r_0$ , spectral (horizontal) peak position  $c_0$ , integrated signal current *a*, spatial width parameters  $\sigma_{\text{spat}}$  and  $w_{\text{spat}}$  and spectral width parameters  $\sigma_{\text{spec}}$  and  $w_{\text{spec}}$ . The variation of the light intensity between frames is removed by normalizing the frame by the integrated signal current:

$$I[r,c] = I_{\text{meas}}[r,c]/a.$$
(3)

The values of the four width parameters are not used, only the fitted peak position  $(r_0, c_0)$  is needed later on."

**30.** Page 8 Line 11 "(across-track and along-track)". This description seems somewhat misleading. The detector has cross-track imaging but the other direction is spatial. Better to leave the "along-track" description out to avoid confusion.

**Not adjusted**, at least not here. The first paragraph of Sect. 5 is very general about terminology, not about the detector. The signal of one read-out does not show the along-track dimension of the PRF, but when consecutive read-outs in flight are concatenated, the two-dimensional shape of the PRF is seen. When the term is used again (page 10, line 13), the text is made clearer. The original text is: "The light in  $\mathbf{K}_{near}$  is considered as the ISRF in the spectral direction and part of the PRF in the spatial direction." The new text is: "... ISRF in the spectral direction and one dimension of the PRF in the spatial direction."

**31.** Page 10 Line 15+ The equations were very difficult to read because of problems with formatting in the PDF. A description of the methodology which is more accessible to a wider audience would be a great addition to the paper. It is left to the authors to ensure that the equations are correct.

Adjusted. We could not see the errors in the PDF on the AMT website. The description of the methodology is perhaps clearer by adding an introduction, where the forward model of adding stray light is given. This also introduces the separate terms in the equations. In the new text, the section starts with: "If **F** represents an ideal frame without stray light, the stable part of the stray light is given by  $\mathbf{K}_{far} \otimes \mathbf{F}$ , where  $\otimes$  indicates a convolution. The integrated signal in the stray light is a fraction  $\sum_{k,l} (\mathbf{K}_{far})_{k,l}$  of the total integrated signal. Measured frame  $\mathbf{J}_0$  is the sum of the ideal frame and the stray light, taking into account that the stray light is removed from the ideal frame:

$$\mathbf{J}_{0} = \left(1 - \sum_{k,l} (\mathbf{K}_{\text{far}})_{k,l}\right) \mathbf{F} + \mathbf{K}_{\text{far}} \otimes \mathbf{F}.$$
(4)

The stable part of the stray light is corrected ... "

32. Page 7 Line 9 Is the 2.1 spread compared to the 1.1 degree source in agreement with the instrument design model? It would be more informative to convert the 1.1 degrees to the expected pixel size. It is just quibbling,nn but the bump reported in the next line looks more like 0.6 or 0.7 %.

**Adjusted.** The spread expressed in pixels and in degrees is indeed as expected. The text is expanded to include this: "The full width at half-maximum of 2.1 spatial pixels corresponds to the stimulus size of about 1.1°, in agreement with the instrument design model."

The bump has a maximum of 0.6 or 0.7 %, but without bump the signal would already be about 0.2 %. The original text implied the difference: "The 0.5 % bump in the laser signal 3 pixels above the line or peak is probably a feature..." The text is changed to: "The signal 3 pixels above the line or peak is 0.5 % higher than the 0.2 % expected from the neighbouring signals. This bump is probably a feature..."

**33.** Page 7 Line 10 The comment concerning the Xe lamp should be explained. It is not clear why once source should spread spatially while the other does not.

**Adjusted.** The reason why the OPO has a bump and the Xe lamp does not is not understood. It is probably due to the setup around the sources. The relevant point is that it is not due to the instrument. To make that clearer, the original text "probably a feature of the stimulus" is expanded to "probably a feature of the stimulus and not the instrument".

34. Page 8 Line 2 "... at the peak positions used ..." "The ISRF was determined ..."

Adjusted according to both suggestions.

35. Page 8 Line 6 "... contain more stray light than any diagonal cross section .." Can electronic contributions be ruled out?

**Adjusted.** Electronic contributions are very unlikely, because far from the peak the horizontal line is slightly rotated (see Fig. 4a) and the vertical line is slightly curved due to spectral smile. The origin of the lines is discussed at the start of Sect. 4, but the deviations from straight lines were not mentioned (for the horizontal line) or only later (for the vertical line). Therefore the text at the start of Sect. 4 is expanded to: "The vertical line through the peak is pure spatial stray light from optics before the spectrometer slit: the common telescope and the relay optics. The line is slightly curved due to the variation of about 0.5 column at a given wavelength but different swath angles, known as the spectral smile. The weak horizontal line through the peak is due to imperfections of the grating line positions. It is slightly rotated, as can be seen at the right side of Fig. 4a."

36. Page 8 Line 8 "... Once the general behaviour of the stray light was examined, ..."

Adjusted according to the suggestion.

37. Page 8 Line 11 "the point response function (PRF)" "point spread function (PSF)" is more commonly used.

**Not adjusted.** In this section, a distinction is made between spread functions and response functions. In those terms, the 'point' function with two spatial dimensions is a response function.

**38.** Page 19 Line 5 "We have developed and applied a novel method ..." While the work presented is thorough and makes use of a very useful algorithm, this may be overstating the situation somewhat...

Adjusted. The word 'novel' is deleted.

**Characterization and correction of stray light in TROPOMI-SWIR**

Paul J. J. Tol1, Tim A. van Kempen1, Richard M. van Hees1, Matthijs Krijger1,2, Sidney Cadot1,3, Ralph Snel1,4, Stefan T. Persijn5, Ilse Aben1, and Ruud W. M. Hoogeveen1 1SRON Netherlands Institute for Space Research, Utrecht, the Netherlands 2Earth Space Solutions, Utrecht, the Netherlands 3Jigsaw B.V., Delft, the Netherlands

4Science and Technology B.V., Delft, the Netherlands

[revised manuscript text omitted]
_{spat}$  and a block distribution in two dimensions  $w_{spat}$  and spectral width parameters  $\sigma_{spec}$  and  $w_{spec}$ . The variation of the light in-

25 tensity is then removed by dividing each between frames is removed by normalizing the frame by the corresponding integrated signal current;

$$\underline{I}[\underline{r,c}] = \underline{I_{\text{meas}}[r,c]/a}.$$
(3)

The values of the four width parameters are not used, only the fitted peak position  $(r_0, c_0)$  is needed later on. In 10 % of the frames, either the peak is too close to the detector edges or one of the fitted widths is too small. Discarding After discarding these cases, there are still 10361 valid frames left.

---

## Author Comment (AC2) · 18 May 2018

*Author comment on* **"Characterization and correction of stray light in TROPOMI-SWIR"** *by* **Paul J. J. Tol et al., manuscript amt-2017-455, Anonymous Referee #2**

We would like to thank Referee #2 for the comments to improve our manuscript. In this document, we provide our reply to the comments. The original comments made by the referee are numbered and typeset in red. Page, line and figure numbers refer to the old version of the manuscript. After the reply we provide a revised version of the manuscript, in which all changes are highlighted.

1. *General Comments:* This paper presents the laboratory characterization of stray light for the TROPOMI SWIR spectrometer, and the development of a correction algorithm to remove the stray light. The stray light is characterized using a monochromatic quasi-point light source; the signal is obtained with more than 7 orders of magnitude by combining saturated and unsaturated measurements taken at different exposure times. Analysis of the spectral-spatial spread function (SSSF), indicates that the stray light can be approximated by the far field of a stable kernel and a main reflection kernel, both of which do not vary with wavelength and across-track position. A correction algorithm is developed to correct stray light in in-flight observations. Application of the stray light correction to stray light measurements as well as synthetic earthshine observations show that the stray light can be significantly reduced. This study presents key information about TROPOMI SWIR and its performance and is very suitable for publication in AMT. This paper is generally well organized, and the methodology and results are generally well described. However, one of the important conclusions "this reduces stray light error sufficiently for accurate gas-column retrievals" is not supported by details. It does not say if the developed correction algorithm is the operational algorithm or will be implemented for operational processing. If it is not the operational algorithm (i.e., described in ATBD). The authors should talk about it in the introduction and if possible compare the performance between this correction algorithm and the operational algorithm. Some of the sentences are difficult to understand and the English writing needs to be improved. The abstract and conclusion can be improved to include more important aspects of this study. Overall, I think that this paper can be published after addressing the comments mentioned here and specific comments below.

   **Adjusted.** Details on the effect of the remaining stray light on retrieval are given in item 29 below. The correction algorithm is the operational algorithm (item 30). Text is clarified with extra explanations and the suggested textual improvements are included (see many items below). Details are added to the abstract and conclusion (items 3, 4, 30 and 31).

   *Specific Comments:*

2. In abstract, L3, the sentence needs to be rephrased to make it read well. You can change it to "For this purpose, it is required to have calibrated radiance measurements in which ..."

   **Adjusted.** The text is changed, also according to a suggestion of Referee #1: "For this purpose, calibrated radiance measurements are needed that are minimally contaminated by instrumental stray light."

3. In abstract, L6, suggest adding "from different exposure times" after "unsaturated measurements"

   **Adjusted.** The text is rephrased as: "The dynamic range of the signal was extended to more than seven orders of magnitude by performing measurements with different exposure times, saturating detector pixels at the longer exposure times."

4. In abstract, L6-8, the sentences do not read well. You may add a little more detail about the algorithm, including the approximation representation with a far-field stable and reflection kernels. Suggest changing the order of sentences to something like "... unsaturated measurements. Analysis of the stray light indicates about 4.4% of the detected light is correctable stray light. An algorithm was then devised ... . Applying this correction significantly reduces the stray-light signal for example ..."

   **Adjusted** according to the suggestion. The new text reads: "Analysis of the stray light indicates about 4.4 % of the detected light is correctable stray light. An algorithm was then devised and implemented in the operational data processor to correct in-flight SWIR observations in near-real time, based on Van Cittert deconvolution. The stray light is

approximated by a far-field kernel independent of position and wavelength and an additional kernel representing the main reflection. Applying this correction significantly reduces the stray-light signal, for example in a simulated dark forest scene close to bright clouds by a factor of about 10."

5. P1, L15, change "distance" to "interval" as it is more frequently used for the spectral dimension

**Adjusted** according to the suggestion.

6. P2, L7, if it is a common telescope, should it be called as "SWIR telescope"? Suggest changing it to "telescope"

**Adjusted.** There are two telescopes: the output of the common telescope goes to the SWIR-only telescope. The text is adjusted: "The SWIR and UVN spectrometers in TROPOMI share a common telescope. After the light is split into the different bands, it is imaged onto the SWIR spectrometer via relay optics and a second telescope."

7. P3, first paragraph, it is not clear about why to avoid saturation at the shortest exposure time, and why fewer frames are averaged at longest exposure time. Based on section 3, the shortest exposure is to measure the strong signal while the longest exposure time is to measure the weakest signal in the wing of point spread function. L3-5 of page 4 can be moved here to help understand the purpose of taking measurement at 4 exposure times.

**Adjusted.** As suggested, the explanation for the different exposure times is included from the next section. Fewer frames are used at the longest exposure time, because fewer are measured due to a lack of calibration time. This is included in the text, which is now: "With a neutral density filter just after the OPO, the spectral radiance at the instrument is reduced to 200 times the highest value in nominal operations. At this radiance, the detector is never saturated at the shortest exposure time. However, the signal with this exposure time is noisy away from the peak. At the three longer exposure times, the signal has a much better signal-to-noise ratio away from the peak, but the peak is saturated. These three exposure times are used to increase the dynamic range of the image by four orders of magnitude (see Sect. 3). At the three shorter exposure times at least 9 frames are averaged, but only 3 frames are taken at the longest exposure time in order to limit the total measurement period."

8. P3, L13-14, change to "ensures that no important features only closer to specific wavelengths or swath angles are missed"

**Adjusted** according to the suggestion.

9. P3, L15, change to "background measurements"

**Adjusted** according to the suggestion.

10. P3, L17, add "as background measurements" after "is used"

**Adjusted** according to the suggestion.

11. P3, L18, suggest changing to "The peak in a given detector area only occurs in a small subset of the light measurements, ..."

**Adjusted** according to the suggestion.

12. P3, L20, suggest "at which" to "and", because "the longest exposure time of 1998 ms", meant by "which" does not occur before "at which"

**Adjusted.** The sentence is split instead. The new text is: "... for this area only data at the longest exposure time of 1998 ms will be used. In that case background measurements are available ..."

13. P4 L6, suggest changing to "... does not saturate a pixel (a saturated signal ...)"

**Adjusted.** The parentheses are included. The original text 'this pixel' and the suggested text 'a pixel' are not needed, because the sentence already starts with the pixel. The new text is: "Then for each pixel, the signal is taken from the longest exposure time that does not saturate (a saturated signal is defined as a signal larger than 90 % of the maximum possible signal)."

14. P4, L8-9, suggest changing to "the signal can be measured with a dynamic range of more than seven orders of magnitude"

Adjusted according to the suggestion, except that 'signal' is replaced by 'signal current'.

15. P4, L11-12, suggest changing to "the signal of unsaturated pixels becomes too high due to spilling from a direct neighbor pixel saturated by light, not dark current"

Adjusted according to the suggestion.

16. P5, L10-11, agree with the first reviewer, the sentence of "The peak positions and ... . Gaussian and a block distribution in these two dimensions ..." is not clear

Adjusted. Block distribution is meant as a more descriptive, alternative name for uniform distribution, but it is clearer to include the fit function explicitly. The original text is: "The peak position and the integrated signal current are determined in each frame by fitting the convolution of a Gaussian and a block distribution in two dimensions. The variation of the light intensity is then removed by dividing each frame by the corresponding integrated signal current." The new text is: "At this stage, a merged frame consists of signal current $I_{\mathrm{meas}}[r, c]$ as a function of detector row $r$ and column $c$. The area near the peak is fitted with a two-dimensional function $S(r, c)$. The one-dimensional convolution of a Gaussian distribution with standard deviation $\sigma$ and a uniform distribution with mean 0 and full width $w$ is given by

$$\mathcal{B}(x; \sigma, w) = \frac{1}{2w} \left[ \mathrm{erf}\left( \frac{x + w/2}{\sqrt{2}\sigma} \right) - \mathrm{erf}\left( \frac{x - w/2}{\sqrt{2}\sigma} \right) \right]. \tag{1}$$

The fit function is

$$S(r, c) = a\, \mathcal{B}(r - r_0; \sigma_{\mathrm{spat}}, w_{\mathrm{spat}})\, \mathcal{B}(c - c_0; \sigma_{\mathrm{spec}}, w_{\mathrm{spec}}), \tag{2}$$

with spatial (vertical) peak position $r_0$, spectral (horizontal) peak position $c_0$, integrated signal current $a$, spatial width parameters $\sigma_{\mathrm{spat}}$ and $w_{\mathrm{spat}}$ and spectral width parameters $\sigma_{\mathrm{spec}}$ and $w_{\mathrm{spec}}$. The variation of the light intensity between frames is removed by normalizing the frame by the integrated signal current:

$$I[r, c] = I_{\mathrm{meas}}[r, c]/a. \tag{3}$$

The values of the four width parameters are not used, only the fitted peak position $(r_0, c_0)$ is needed later on."

17. P5, L13, change to "After discarding"

Adjusted according to the suggestion.

18. P5, last paragraph in Sect. 3, do the last four sentences describe the PRNU correction as it is not clear to me. If not, it is useful to describe the PRNU in a few sentences.

Adjusted. The last four sentences do not describe the PRNU correction. A description is added: "After these steps, each merged frame is corrected for pixel response non-uniformity (PRNU). This is the pixel-to-pixel variation of the gain (ratio of the signal current to the input photon rate), determined in separate calibration measurements using a light source with a flat spectrum. This correction is very small, up to about 1 %." The following text forms a new paragraph.

19. P9, L2-3, it is not clear why only stray light originating at spatial coordinates is corrected. Does it mean that the spectral stray light is not corrected? Please clarify it.

Adjusted. The stress in the sentence is not clear. The spatial coordinates refer to the source external to the instrument, as long as it is in view and not just outside it. Once it is in view, the full stray light is corrected. The new text describes it differently: "Stray light is only corrected when the source itself is imaged on the detector, because the direct source light serves as input for the correction."

20. P10, L4, change "using the element-wise product o" to "where o is the element-wise product"

Adjusted according to the suggestion.

21. P10, L21, add "is as follows" after "The rationale of this algorithm" as it is not a sentence and change "which is" to "and is"

**Adjusted** according to both suggestions.

22. P11, L1, change to "... center. The iteration consists of ..."

**Adjusted** according to the suggestion.

23. P11, L7, change to ", as explained"

**Adjusted** according to the suggestion.

24. P15, L8, suggest adding "After the stray light correction" before "Some stray light ..." to make it more readable.

**Adjusted.** A whole sentence is added instead, in response to a question of Referee #1. The new text is: "In these cases with illumination over many pixels and at regular intensities, the stray-light correction works well. Some stray light remains..."

25. P16, Figure 14 caption, change "measurement" to "measurements"

**Adjusted** according to the suggestion.

26. P16, the last paragraph that continues on P18 is not easy to understand and seems to distract the flow from Fig. 15a to Fig. 15b. I think that the main purpose is to describe how to construct signal with stray light by convolving Fig 15 a with the SSSF. It is not clear to me about the need to talk about merged frame and conversion to kernels here. Please make the description clearer.

**Adjusted.** For an informative stray-light simulation, the SSSF cannot be the stable kernel. It has to be based on the original merged frames, but these have to be normalized first. An introduction is added to the start of the paragraph. That part becomes: "Stray light is introduced by convolving the expected signal with the SSSF. However, if the stable kernel would be used as an SSSF approximation for the whole detector, the effect of neglected features moving relative to the peak cannot be assessed. Instead, the detector is divided into areas of pixels around all peak positions found in the merged frames. A given measured merged frame constitutes the most accurate SSSF available for the detector pixels in the corresponding area. Care has to be taken with signal scaling and padding to the right dimensions before a merged frame can be used in a convolution."

The procedure that follows can be reduced by referring to an introduction that is added to the start of Sect. 6. The new procedure text is: "A merged frame is converted to a kernel by applying the same normalization factor as in the creation of the stable kernel and padding to the same dimensions with the peak at the centre. The padding values are the values in the stable kernel at the same positions. This is followed by multiplication with $\mathbf{M}_{far}$. The resulting kernel is used as $\mathbf{K}_{far}$ in Eq. (6) and applied to the corresponding area in the simulated radiance frame. This is repeated for all merged frames and corresponding areas in the radiance frame and the sum is taken."

The actual application of the kernel is no longer described in words here, but given with an equation at the start of Sect. 6 in extra text: "If $\mathbf{F}$ represents an ideal frame without stray light, the stable part of the stray light is given by $\mathbf{K}_{far} \otimes \mathbf{F}$, where $\otimes$ indicates a convolution. The integrated signal in the stray light is a fraction $\sum_{k,l}(\mathbf{K}_{far})_{k,l}$ of the total integrated signal. Measured frame $\mathbf{J}_0$ is the sum of the ideal frame and the stray light, taking into account that the stray light is removed from the ideal frame:

$$\mathbf{J}_0 = \left(1 - \sum_{k,l}(\mathbf{K}_{far})_{k,l}\right)\mathbf{F} + \mathbf{K}_{far} \otimes \mathbf{F}.$$

The stable part of the stray light..."

27. Figure 16 caption, suggest changing to "... respectively, for the forest spectrum in 2305-2385 nm located four rows from the cloud spectra"

**Adjusted** according to the suggestion, but adding back 'performance range'. The new text is: "..., respectively, for the forest spectrum in performance range 2305–2385 nm located four rows from the cloud spectra."

28. P18, L9-10, it is not clear about "one merged frame per 20 pixels". Do you mean merging 20 pixels across the track (i.e., merging 20 rows)?

**Adjusted.** We did not mean 20 ground pixels but a basically rectangular area of detector pixels. The new text is: "...there is only one merged frame per area of about $3 \times 8$ detector pixels." The numerical difference is due to rounding.

29. P19, can you provide more quantitative details to support the last sentence of section 8 as this is an important conclusion in abstract and conclusion and the residual error is still up to 1%. In addition, has the instrument noise been added to the simulation? Will the inclusion of instrument noise affect the performance of stray light correction?

**Adjusted.** A full retrieval analysis with statistics over different scenes using the final retrieval algorithm is beyond the scope of this article. The given conclusion is based on one challenging scene type and a non-scattering retrieval algorithm. The old text is: "Calculations have shown that the error in the retrieved gas columns is within the allowed error budget." This is replaced by more detailed information: "Applying non-scattering retrieval (without aerosol or cirrus parameters) to the forest spectrum used for Fig. 16, the error of the methane column reduces from 14 % before correction to 0.26 % after correction, within the error budget of 0.35 % for stray light. The error of the CO column reduces from 26 % before correction to 1.3 % after correction, within the stray-light error budget for the given CO column of 3.0 %. The forest spectra at other rows give similar results after correction."

The retrieval errors (and the error budgets) are given for simulations with noise only due to the measured merged frames used as SSSF. Simulations with read noise and shot noise added before stray-light correction and retrieval show that the total retrieval error is the quadratic sum of the errors due to noise and corrected stray-light separately. Therefore the noise does not affect the performance of the stray-light correction.

30. Also, it is not clear if this stray light correction algorithm is the operational algorithm to be used in level 0 to 1b processor (i.e., described in ATBD)? It looks like it is not, then how is the performance and speed of this correction compared to those of the operational correction?

**Adjusted.** This is the operational algorithm for stray-light correction used in the level 0 to 1b processor, but only for SWIR data. The ATBD describes it very sketchy and somewhat outdated in Sect. 24.1.6. Text is added in the abstract and conclusion. In the abstract, the old text was: "An algorithm was then devised to correct in-flight observations...". This is replaced by: "An algorithm was then devised and implemented in the operational data processor to correct in-flight SWIR observations..." In the conclusions, a shorter variation is given: "A fast correction algorithm has been devised and implemented in the operational data processor, based on Van Cittert deconvolution."

31. In conclusion, it is useful to expand a little bit about the main idea of the fast correction algorithm (e.g., stable kernel, reflection kernel etc.)

**Adjusted.** In Sect. 9 with the conclusions, an extra sentence is added after the mention of the Van Cittert deconvolution: "The stray light is approximated by a far-field kernel independent of position and wavelength and an additional kernel representing the main reflection."

32. P20, L2, this sentence does not read well, suggest changing to "The fitted r_center is 127.5, exactly ..."

**Adjusted.** The value of $r_{\text{centre}}$ is not fitted but chosen. The new text is: "...
[revised manuscript text omitted]

---

## Author Comment (AC3) · 18 May 2018

*Author comment on* **"Characterization and correction of stray light in TROPOMI-SWIR"** *by* **Paul J. J. Tol et al., manuscript amt-2017-455, Anonymous Referee #3**

We would like to thank Referee #3 for the comments to improve our manuscript. In this document, we provide our reply to the comments. The original comments made by the referee are numbered and typeset in red. Page, line and figure numbers refer to the old version of the manuscript. After the reply we provide a revised version of the manuscript, in which all changes are highlighted.

*General Comments. Initial paragraph or section evaluating the overall quality of the discussion paper. The paper is well written and of good quality, with a considerable number of new interesting topics and techniques related to stray light characterization and correction, and shall certainly be published. However, I am of the opinion that the quality of the paper can be much improved to be more useful with a comparatively small additional effort, in line with the comments and suggestions provided below. After these comments and suggestions have been adequately addressed, the paper shall certainly be published.*

1. *For the applications in TROPOMI SWIR it is essential to express, quantify and present stray light at L0 and L1b as a percentage of the useful signal for realistic earth atmosphere low-albedo scenes and signals within absorption peaks. This is currently not the case. For example in figure 14 it can be seen that the difference between bright and dark scenes is roughly a factor 8 (see also page 15, line 14), and the difference between rather deep absorption line and continuum is also roughly about a factor 8. Both together would make a difference of a factor 64. For example, in the legend of figure 15 (but also in other places in the paper) it is clear that the stray light is expressed as percentage of the expected continuum in the given row, not as percentage of the useful signal in the absorption lines. If the authors prefer to present the stray light like this, it is essential to at least also show the stray light as percentage with respect to the low albedo useful signal in the absorption lines. This shall be added to the paper in the text, figures and conclusions.*

   **Not adjusted.** The stray light was expressed as a percentage of the continuum after consultation of the team that does the operational trace-gas retrieval in SWIR. The reasoning can be explained as follows. In Fig. 14, the signal in the deepest absorption line is 6 % of the continuum in the cloud spectrum and only 1 % of the continuum in the forest spectrum, due to the longer air column to the ground. The used U.S. Standard atmosphere will be drier than the atmosphere in many other scenes, so the signal can be even lower. If the stray light would be expressed as a percentage of that signal, the values would be far above 100 % and extremely dependent on the exact assumptions about the atmospheric composition. This would render the values meaningless. Therefore, the stray light is compared with the continuum (of the low-albedo spectrum), i.e. it is seen as an absolute signal contribution instead of a relative one. Then it turns out to be basically flat before and after correction (Fig. 16), so it is easier to see deviations and to compare spectra. A plot of the stray light relative to the expected signal at the same wavelength would look like an inverted radiance spectrum. Showing the absolute stray light is also in line with operational methane and CO retrieval, where the absolute difference between measurement and model is minimized.

   *Specific comments. Section addressing individual scientific questions/issues.*

2. *Page 8, figure 7: It is recommended to (also) show the row distances similarly as the column distances as shown in figure 8, to allow a better comparison between the two. For example, it is recommended to add a figure like the lower figure in figure 7 at a horizontal scale of +/-100 pixels, to allow better comparison with figure 8.*

   **Not adjusted.** The cross sections in Figs. 7 and 8 use a limited data set, at all swath angles but only one wavelength, to show the peak area before it is averaged over wavelength to create the stable kernel. The main reflection is always somewhere on the vertical cross section, but at every swath angle at a different position. To see the average cross section outside the reflection over a large row range, the data bins have to be enlarged from 0.025 pixel to 0.5 pixel. The result is shown in Fig. 1 in this author comment. The reflection is still visible at $-90$ and $-14$ rows as bumps and at $+85$ rows as a second curve, formed by every second point. Other deviations from a smooth curve may also be remnants of the reflection, so the data are difficult to interpret. The different bins in the linear and logarithmic plot would also need explaining. Both points would distract from the main focus of the article. Therefore only the central part of the

[Figure]

**Figure 1.** Vertical cross section through the peak using the combined data at 2377.90 nm on a logarithmic scale, but on a larger range than in the article. Included is a fit with the convolution of a Gaussian and a uniform distribution.

cross section is given in the article, with the same bins as in the linear plot. A larger range near the peak is given in the two-dimensional colour plots of Fig. 6 and the full range is shown in the stable-kernel plots of Fig. 9.

3. Page 11, line 20: The number is given as 4.3% of the detected light. In line with the general comment given above, it is recommended to provide also the percentage numbers with respect to low-albedo numbers in the absorption lines.

   **Not adjusted.** The 4.3 % is a property of the far-field kernel, independent of the scene to which it is applied. This section describes the calibration data, among which is the far-field kernel. Application to scenes is the topic of the next section. See item 1 regarding the general point on percentages.

4. Page 18, lines 7-10: Again, in line with the above, this is why it is important to calculate the stray light fraction with respect to the useful signal, because the absorption lines and low-albedo scenes will be filled in with higher signals from the continuum and the higher-albedo scenes. This may also affect the instrument spectral response functions. Please reflect this in the text.

   **Not adjusted.** Regarding the calculation of the stray-light fraction, see item 1. The far-field kernel describing the main stray light and the instrument spectral response function are two complementary parts of the same stable kernel, which is a property of the instrument and is not affected by the scenes.

5. Page 18, lines 11-12: Again, in line with the above, this is as written not agreed. This sentence shall be removed. In case the authors want to keep this sentence, another sentence needs to be added explaining why stray light needs to be assessed as fraction with respect to the useful signal, not the signal in the continuum.

   **Not adjusted.** See item 1.

6. Section 9, conclusions. The statement "It is expected that this brings the stray-light error in gas-column retrievals within the required budget" is not discussed or supported by analyses in the paper. Either provide more support material in the paper to corroborate this statement, or remove it from the conclusions.

   **Adjusted.** A full retrieval analysis with statistics over different scenes using the final retrieval algorithm is beyond the scope of this article. The given conclusion is based on one challenging scene type and a non-scattering retrieval algorithm. At the end of Sect. 8 extra information is given. The old text in Sect. 8 is: "Calculations have shown that the error in the retrieved gas columns is within the allowed error budget." This is replaced by: "Applying non-scattering retrieval (without aerosol or cirrus parameters) to the forest spectrum used for Fig. 16, the error of the methane column reduces from 14 % before correction to 0.26 % after correction, within the error budget of 0.35 % for stray light. The error of the CO column reduces from 26 % before correction to 1.3 % after correction, within the stray-light error budget

for the given CO column of 3.0 %. The forest spectra at other rows give similar results after correction." The conclusion changes from old "It is expected that ..." to new "Simulations indicate that ...".

*Technical corrections. Compact listing of purely technical corrections (typing errors, etc.).*

7. The legends of (almost) all figures have symbols that cannot be read. This is at least true in the pdf version. Please correct.

   **Not adjusted yet.** We could not see the errors in the PDF on the AMT website, using several PDF viewers in several operating systems. That makes it very difficult to implement effective changes. We will act upon instructions given by the Editor.

8. (Almost) all equations (e.g. 3,4,5,7,8,9,10) have symbols that cannot be read. This is at least true in the pdf version. Please correct.

   **Not adjusted yet.** The document has been generated with pdflatex in the standard way and all fonts are embedded. We could not see the errors in the PDF on the AMT website, using several PDF viewers in several operating systems. We will act upon instructions given by the Editor.

9. Figure 15: In the legends, please indicate if this is a fraction, percentage or something else.

   **Not adjusted.** Subplots (a) and (b) are absolute currents in femtoampere, as given in the legend. Subplots (c) and (d) are "expressed as a percentage of the expected continuum in the given row" as written in the caption. This is too long to put in the legend, which uses "normalized [%]" instead. Perhaps this was not clear due to item 7.

[revised manuscript text omitted]

---

## Referee Report (RR1)

*Characterization and correction of stray light in TROPOMI-SWIR, Paul J.J. Tol et al., MS No.: amt-2017-455,*

**General comments**

Initial paragraph or section evaluating the overall quality of the discussion paper.

The paper is well written and of good quality, with a considerable number of new interesting topics and techniques related to stray light characterization and correction, and shall certainly be published. However, I am of the opinion that the quality of the paper can be much improved to be more useful with a comparatively small additional effort, in line with the comments and suggestions provided below. After these comments and suggestions have been adequately addressed, the paper shall certainly be published.

1. For the applications in TROPOMI SWIR it is essential to express, quantify and present stray light at L0 and L1b as a percentage of the useful signal for realistic earth atmosphere low-albedo scenes and signals within absorption peaks. This is currently not the case. For example in figure 14 it can be seen that the difference between bright and dark scenes is roughly a factor 8 (see also page 15, line 14), and the difference between rather deep absorption line and continuum is also roughly about a factor 8. Both together would make a difference of a factor 64. For example, in the legend of figure 15 (but also in other places in the paper) it is clear that the stray light is expressed as percentage of the expected continuum in the given row, not as percentage of the useful signal in the absorption lines. If the authors prefer to present the stray light like this, it is essential to at least also show the stray light as percentage with respect to the low albedo useful signal in the absorption lines. This shall be added to the paper in the text, figures and conclusions.

**Not adjusted.** The stray light was expressed as a percentage of the continuum after consultation of the team that does the operational trace-gas retrieval in SWIR. The reasoning can be explained as follows. In Fig. 14, the signal in the deepest absorption line is 6 % of the continuum in the cloud spectrum and only 1 % of the continuum in the forest spectrum, due to the longer air column to the ground. The used U.S. Standard atmosphere will be drier than the atmosphere in many other scenes, so the signal can be even lower. If the stray light would be expressed as a percentage of that signal, the values would be far above 100 % and extremely dependent on the exact assumptions about the atmospheric composition.
This would render the values meaningless. Therefore, the stray light is compared with the continuum (of the low-albedo spectrum), i.e. it is seen as an absolute signal contribution instead of a relative one. Then it turns out to be basically flat before and after correction (Fig. 16), so it is easier to see deviations and to compare spectra. A plot of the stray light relative to the expected signal at the same wavelength would look like an inverted radiance spectrum. Showing the absolute stray light is also in line with operational methane and CO retrieval, where the absolute difference between measurement and model is minimized.

Response 19 June 2018:
I do not agree with this response. As pointed out in the original comment I wasn't asking to change the approach, but I was merely asking for an additional figure showing the stray light in a different, in my view more meaningful, way. I do not agree that it makes sense to present percentual stray light fractions with respect to a signal obtained at a different spatial and/or spectral location on the detector. For me the only meaningful parameter is the percentual stray light contribution calculated with respect to the appropriate useful signal at the same spatial and spectral location on the detector. I took note of the authors' comments that scientifically it makes more sense to calculate the stray light fraction with respect to the continuum, but I do not agree with that statement. I agree that calculating stray light with respect to the useful signal at the same spectral and spatial location on the detector increases the stray light percentage quite considerably (which is exactly the point, actually) and makes the figure(s) look more variable (which is also actually the point), but I strongly disagree with the author's statement that this "would render the values meaningless".
As a compromise I offered to allow the authors to present the figures in their preferred way, but add one (or some) additional figure(s) showing the stray light percentages calculated with respect to useful signals at the same spatial and spectral location on the detector at which the stray light is calculated. However, the authors did not yet accept this compromise. At the current stage, having taken note of the author's response, it is my opinion that it is important to add one or more of the figures as requested by

me above and in the original comment, because to my opinion the figures showing stray light at a particular location with respect to a signal at another spatial/spectral location is not meaningful and presents an excessively overly optimistic view on the instrument stray light situation at Level-0 and Level-1b. Therefore I recommend to the editors that one or more figures are added to the paper, as requested by me in the above and earlier comment, prior to publication.

**Specific comments**

Section addressing individual scientific questions/issues.

2.
Page 8, figure 7:
It is recommended to (also) show the row distances similarly as the column distances as shown in figure 8, to allow a better comparison between the two. For example, it is recommended to add a figure like the lower figure in figure 7 at a horizontal scale of +/-100 pixels, to allow better comparison with figure 8.

**Not adjusted.** The cross sections in Figs. 7 and 8 use a limited data set, at all swath angles but only one wavelength, to show the peak area before it is averaged over wavelength to create the stable kernel. The main reflection is always somewhere on the vertical cross section, but at every swath angle at a different position. To see the average cross section outside the reflection over a large row range, the data bins have to be enlarged from 0.025 pixel to 0.5 pixel. The result is shown in Fig. 1 in this author comment. The reflection is still visible at −90 and −14 rows as bumps and at +85 rows as a second curve, formed by every second point. Other deviations from a smooth curve may also be remnants of the reflection, so the data are difficult to interpret. The different bins in the linear and logarithmic plot would also need explaining. Both points would distract from the main focus of the article. Therefore only the central part of the cross section is given in the article, with the same bins as in the linear plot. A larger range near the peak is given in the two-dimensional colour plots of Fig. 6 and the full range is shown in the stable-kernel plots of Fig. 9.

Response 19 June 2018:
The reason for asking the cross-sections as figure examples in both spectral and spatial dimensions is that the cumulative stray light is originating from the whole illuminated image on the detector, i.e. by summing all contributions from the illuminated detector into the region where the stray light is calculated, and therefore it makes sense to add the figure as requested in the original comment, to show an example of the relative intensities in spectral and spatial dimensions on comparable scales.
I therefore recommend to the editors that the requested figure with the larger scale is added to the paper, as requested in the comment, prior to publication. This is not a big change to the approach of the document (which is not disputed), but for me it adds meaningful and significant essential information to the paper.

3.
Page 11, line 20:
The number is given as 4.3% of the detected light.
In line with the general comment given above, it is recommended to provide also the percentage numbers with respect to low-albedo numbers in the absorption lines.

**Not adjusted.** The 4.3 % is a property of the far-field kernel, independent of the scene to which it is applied. This section describes the calibration data, among which is the far-field kernel. Application to scenes is the topic of the next section.
See item 1 regarding the general point on percentages.

Response 19 June 2018:
See response to point #1.

4.
Page 18, lines 7-10:
Again, in line with the above, this is why it is important to calculate the stray light fraction with respect to the useful signal, because the absorption lines and low-albedo scenes will be filled in with higher

signals from the continuum and the higher-albedo scenes. This may also affect the instrument spectral response functions. Please reflect this in the text.

**Not adjusted.** Regarding the calculation of the stray-light fraction, see item 1. The far-field kernel describing the main stray light and the instrument spectral response function are two complementary parts of the same stable kernel, which is a property of the instrument and is not affected by the scenes.

Response 19 June 2018: See response to point #1.

5.
Page 18, lines 11-12:
Again, in line with the above, this is as written not agreed. This sentence shall be removed. In case the authors want to keep this sentence, another sentence needs to be added explaining why stray light needs to be assessed as fraction with respect to the useful signal, not the signal in the continuum.

**Not adjusted.** See item 1.

Response 19 June 2018: See response to point #1.

6.
Section 9, conclusions.
The statement "It is expected that this brings the stray-light error in gas-column retrievals within the required budget" is not discussed or supported by analyses in the paper. Either provide more support material in the paper to corroborate this statement, or remove it from the conclusions.

**Adjusted.** A full retrieval analysis with statistics over different scenes using the final retrieval algorithm is beyond the scope of this article. The given conclusion is based on one challenging scene type and a non-scattering retrieval algorithm. At the end of Sect. 8 extra information is given. The old text in Sect. 8 is: "Calculations have shown that the error in the retrieved gas columns is within the allowed error budget." This is replaced by: "Applying non-scattering retrieval (without aerosol or cirrus parameters) to the forest spectrum used for Fig. 16, the error of the methane column reduces from 14 % before correction to 0.26 % after correction, within the error budget of 0.35 % for stray light. The error of the CO column reduces from 26 % before correction to 1.3 % after correction, within the stray-light error budget 2 for the given CO column of 3.0 %. The forest spectra at other rows give similar results after correction." The conclusion changes from old "It is expected that ..." to new "Simulations indicate that ...".

Response 19 June 2018: Okay for this paper, although I am of the opinion that these statements cannot be corroborated by the material presented in this paper, and should be deferred to another paper where more details are given on the analyses / simulations and their conclusions, because currently, within the scope of this paper, the numbers and statements cannot be checked or verified..

**Technical corrections**

Compact listing of purely technical corrections (typing errors, etc.).

7.
The legends of (almost) all figures have symbols that cannot be read. This is at least true in the pdf version. Please correct.

**Not adjusted yet.** We could not see the errors in the PDF on the AMT website, using several PDF viewers in several operating systems. That makes it very difficult to implement effective changes. We will act upon instructions given by the Editor.

Response 19 June 2018:
I am not sure what happened. It looks okay now, also on my side. Maybe something strange happens when printing?
Comment withdrawn, is okay now.

8.

(Almost) all equations (e.g. 3,4,5,7,8,9,10) have symbols that cannot be read. This is at least true in the pdf version. Please correct.

**Not adjusted yet.** The document has been generated with pdflatex in the standard way and all fonts are embedded. We could not see the errors in the PDF on the AMT website, using several PDF viewers in several operating systems. We will act upon instructions given by the Editor.

Response 19 June 2018:
I am not sure what happened. It looks okay now, also on my side. Maybe something strange happens when printing?
Comment withdrawn, is okay now.

9.
Figure 15: In the legends, please indicate if this is a fraction, percentage or something else.

**Not adjusted.** Subplots (a) and (b) are absolute currents in femtoampere, as given in the legend. Subplots (c) and (d) are "expressed as a percentage of the expected continuum in the given row" as written in the caption. This is too long to put in the legend, which uses "normalized [%]" instead. Perhaps this was not clear due to item 7.

Response 19 June 2018:
Looks okay now also on my side, indeed probably from point 7, comment withdrawn.

---

## Author Response (AR3)

*Author comment on* **"Characterization and correction of stray light in TROPOMI-SWIR"** *by* **Paul J. J. Tol et al., manuscript amt-2017-455, Anonymous Referee #3**

We would like to thank Referee #3 for the comments to improve our manuscript. In this document, we provide our reply to the comments. The original comments made by the referee are numbered and typeset in red. Page, line and figure numbers refer to the old version of the manuscript. After the reply we provide a revised version of the manuscript, in which all changes are highlighted.

*General Comments. Initial paragraph or section evaluating the overall quality of the discussion paper. The paper is well written and of good quality, with a considerable number of new interesting topics and techniques related to stray light characterization and correction, and shall certainly be published. However, I am of the opinion that the quality of the paper can be much improved to be more useful with a comparatively small additional effort, in line with the comments and suggestions provided below. After these comments and suggestions have been adequately addressed, the paper shall certainly be published.*

1. *For the applications in TROPOMI SWIR it is essential to express, quantify and present stray light at L0 and L1b as a percentage of the useful signal for realistic earth atmosphere low-albedo scenes and signals within absorption peaks. This is currently not the case. For example in figure 14 it can be seen that the difference between bright and dark scenes is roughly a factor 8 (see also page 15, line 14), and the difference between rather deep absorption line and continuum is also roughly about a factor 8. Both together would make a difference of a factor 64. For example, in the legend of figure 15 (but also in other places in the paper) it is clear that the stray light is expressed as percentage of the expected continuum in the given row, not as percentage of the useful signal in the absorption lines. If the authors prefer to present the stray light like this, it is essential to at least also show the stray light as percentage with respect to the low albedo useful signal in the absorption lines. This shall be added to the paper in the text, figures and conclusions.*

   **Not adjusted.** The stray light was expressed as a percentage of the continuum after consultation of the team that does the operational trace-gas retrieval in SWIR. The reasoning can be explained as follows. In Fig. 14, the signal in the deepest absorption line is 6 % of the continuum in the cloud spectrum and only 1 % of the continuum in the forest spectrum, due to the longer air column to the ground. The used U.S. Standard atmosphere will be drier than the atmosphere in many other scenes, so the signal can be even lower. If the stray light would be expressed as a percentage of that signal, the values would be far above 100 % and extremely dependent on the exact assumptions about the atmospheric composition. This would render the values meaningless. Therefore, the stray light is compared with the continuum (of the low-albedo spectrum), i.e. it is seen as an absolute signal contribution instead of a relative one. Then it turns out to be basically flat before and after correction (Fig. 16), so it is easier to see deviations and to compare spectra. A plot of the stray light relative to the expected signal at the same wavelength would look like an inverted radiance spectrum. Showing the absolute stray light is also in line with operational methane and CO retrieval, where the absolute difference between measurement and model is minimized.

   Response 19 June 2018: I do not agree with this response. As pointed out in the original comment I wasn't asking to change the approach, but I was merely asking for an additional figure showing the stray light in a different, in my view more meaningful, way. I do not agree that it makes sense to present percentual stray light fractions with respect to a signal obtained at a different spatial and/or spectral location on the detector. For me the only meaningful parameter is the percentual stray light contribution calculated with respect to the appropriate useful signal at the same spatial and spectral location on the detector. I took note of the authors' comments that scientifically it makes more sense to calculate the stray light fraction with respect to the continuum, but I do not agree with that statement. I agree that calculating stray light with respect to the useful signal at the same spectral and spatial location on the detector increases the stray light percentage quite considerably (which is exactly the point, actually) and makes the figure(s) look more variable (which is also actually the point), but I strongly disagree with the author's statement that this "would render the values meaningless".

   As a compromise I offered to allow the authors to present the figures in their preferred way, but add one (or some) additional figure(s) showing the stray light percentages calculated with respect to useful signals at the same spatial

[Figure]

**Figure 1.** Vertical cross section through the peak using the combined data at 2377.90 nm on a logarithmic scale, but on a larger range than in the article. Included is a fit with the convolution of a Gaussian and a uniform distribution.

and spectral location on the detector at which the stray light is calculated. However, the authors did not yet accept this compromise. At the current stage, having taken note of the author's response, it is my opinion that it is important to add one or more of the figures as requested by me above and in the original comment, because to my opinion the figures showing stray light at a particular location with respect to a signal at another spatial/spectral location is not meaningful and presents an excessively overly optimistic view on the instrument stray light situation at Level-0 and Level-1b. Therefore I recommend to the editors that one or more figures are added to the paper, as requested by me in the above and earlier comment, prior to publication.

Response 12 July 2018: **Adjusted.** As requested, two figures have been added where the stray light is given with respect to the useful signal at the same location on the detector: new Fig. 18 with detector maps before and after correction and new Fig. 20 with cross sections of these maps. Descriptive text has also been added. For a better text flow, previous Fig. 16 with four panels has been split into new Figs. 17 and 19 with two panels each.

*Specific comments. Section addressing individual scientific questions/issues.*

2. Page 8, figure 7: It is recommended to (also) show the row distances similarly as the column distances as shown in figure 8, to allow a better comparison between the two. For example, it is recommended to add a figure like the lower figure in figure 7 at a horizontal scale of +/-100 pixels, to allow better comparison with figure 8.

**Not adjusted.** The cross sections in Figs. 7 and 8 use a limited data set, at all swath angles but only one wavelength, to show the peak area before it is averaged over wavelength to create the stable kernel. The main reflection is always somewhere on the vertical cross section, but at every swath angle at a different position. To see the average cross section outside the reflection over a large row range, the data bins have to be enlarged from 0.025 pixel to 0.5 pixel. The result is shown in Fig. 1 in this author comment. The reflection is still visible at $-90$ and $-14$ rows as bumps and at $+85$ rows as a second curve, formed by every second point. Other deviations from a smooth curve may also be remnants of the reflection, so the data are difficult to interpret. The different bins in the linear and logarithmic plot would also need explaining. Both points would distract from the main focus of the article. Therefore only the central part of the cross section is given in the article, with the same bins as in the linear plot. A larger range near the peak is given in the two-dimensional colour plots of Fig. 6 and the full range is shown in the stable-kernel plots of Fig. 9.

Response 19 June 2018: The reason for asking the cross-sections as figure examples in both spectral and spatial dimensions is that the cumulative stray light is originating from the whole illuminated image on the detector, i.e. by summing all contributions from the illuminated detector into the region where the stray light is calculated, and therefore it makes sense to add the figure as requested in the original comment, to show an example of the relative intensities in spectral and

spatial dimensions on comparable scales. I therefore recommend to the editors that the requested figure with the larger scale is added to the paper, as requested in the comment, prior to publication. This is not a big change to the approach of the document (which is not disputed), but for me it adds meaningful and significant essential information to the paper.

Response 12 July 2018: **Adjusted.** A figure has been added with the spatial cross section as shown in Fig. 1 in this author comment, but including two extra lines. The reason for asking the figure was to compare the relative intensities in spectral and spatial dimensions. Therefore the spectral cross section is repeated. These two cross sections are special cases with more stray light than in any other cross section. Therefore a third line shows a diagonal cross section.

3. Page 11, line 20: The number is given as 4.3% of the detected light. In line with the general comment given above, it is recommended to provide also the percentage numbers with respect to low-albedo numbers in the absorption lines.

   **Not adjusted.** The 4.3 % is a property of the far-field kernel, independent of the scene to which it is applied. This section describes the calibration data, among which is the far-field kernel. Application to scenes is the topic of the next section. See item 1 regarding the general point on percentages.

   Response 19 June 2018: See response to point #1.

   Response 12 July 2018: **Adjusted.** Figures and text have been added to the article, described in the response in item 1.

4. Page 18, lines 7-10: Again, in line with the above, this is why it is important to calculate the stray light fraction with respect to the useful signal, because the absorption lines and low-albedo scenes will be filled in with higher signals from the continuum and the higher-albedo scenes. This may also affect the instrument spectral response functions. Please reflect this in the text.

   **Not adjusted.** Regarding the calculation of the stray-light fraction, see item 1. The far-field kernel describing the main stray light and the instrument spectral response function are two complementary parts of the same stable kernel, which is a property of the instrument and is not affected by the scenes.

   Response 19 June 2018: See response to point #1.

   Response 12 July 2018: **Adjusted.** Figures and text have been added to the article, described in the response in item 1.

5. Page 18, lines 11-12: Again, in line with the above, this is as written not agreed. This sentence shall be removed. In case the authors want to keep this sentence, another sentence needs to be added explaining why stray light needs to be assessed as fraction with respect to the useful signal, not the signal in the continuum.

   **Not adjusted.** See item 1.

   Response 19 June 2018: See response to point #1.

   Response 12 July 2018: **Adjusted.** Figures and text have been added to the article, described in the response in item 1.

6. Section 9, conclusions. The statement "It is expected that this brings the stray-light error in gas-column retrievals within the required budget" is not discussed or supported by analyses in the paper. Either provide more support material in the paper to corroborate this statement, or remove it from the conclusions.

   **Adjusted.** A full retrieval analysis with statistics over different scenes using the final retrieval algorithm is beyond the scope of this article. The given conclusion is based on one challenging scene type and a non-scattering retrieval algorithm. At the end of Sect. 8 extra information is given. The old text in Sect. 8 is: "Calculations have shown that the error in the retrieved gas columns is within the allowed error budget." This is replaced by: "Applying non-scattering retrieval (without aerosol or cirrus parameters) to the forest spectrum used for Fig. 16, the error of the methane column reduces from 14 % before correction to 0.26 % after correction, within the error budget of 0.35 % for stray light. The error of the CO column reduces from 26 % before correction to 1.3 % after correction, within the stray-light error budget for the given CO column of 3.0 %. The forest spectra at other rows give similar results after correction." The conclusion changes from old "It is expected that ..." to new "Simulations indicate that ...".

   Response 19 June 2018: Okay for this paper, although I am of the opinion that these statements cannot be corroborated by the material presented in this paper, and should be deferred to another paper where more details are given on the analyses

/ simulations and their conclusions, because currently, within the scope of this paper, the numbers and statements cannot be checked or verified.

Response 12 July 2018: **Agreed.** Okay for this paper.

*Technical corrections. Compact listing of purely technical corrections (typing errors, etc.).*

7. The legends of (almost) all figures have symbols that cannot be read. This is at least true in the pdf version. Please correct.

   **Not adjusted yet.** We could not see the errors in the PDF on the AMT website, using several PDF viewers in several operating systems. That makes it very difficult to implement effective changes. We will act upon instructions given by the Editor.

   Response 19 June 2018: I am not sure what happened. It looks okay now, also on my side. Maybe something strange happens when printing? Comment withdrawn, is okay now.

8. (Almost) all equations (e.g. 3,4,5,7,8,9,10) have symbols that cannot be read. This is at least true in the pdf version. Please correct.

   **Not adjusted yet.** The document has been generated with pdflatex in the standard way and all fonts are embedded. We could not see the errors in the PDF on the AMT website, using several PDF viewers in several operating systems. We will act upon instructions given by the Editor.

   Response 19 June 2018: I am not sure what happened. It looks okay now, also on my side. Maybe something strange happens when printing? Comment withdrawn, is okay now.

9. Figure 15: In the legends, please indicate if this is a fraction, percentage or something else.

   **Not adjusted.** Subplots (a) and (b) are absolute currents in femtoampere, as given in the legend. Subplots (c) and (d) are "expressed as a percentage of the expected continuum in the given row" as written in the caption. This is too long to put in the legend, which uses "normalized [%]" instead. Perhaps this was not clear due to item 7.

   Response 19 June 2018: Looks okay now also on my side, indeed probably from point 7, comment withdrawn.

[revised manuscript text omitted]